# Targeting EZH2 reactivates a breast cancer subtype-specific anti-metastatic transcriptional program

Alison Hirukawa[1,2], Harvey W. Smith[1], Dongmei Zuo[1], Catherine R. Dufour[1], Paul Savage[1], Nicholas Bertos[1], Radia M. Johnson[1], Tung Bui[1,2], Guillaume Bourque[3,4], Mark Basik[5,6], Vincent Giguère[1,2,6], Morag Park[1,2,6] & William J. Muller[1,2,6]

Emerging evidence has illustrated the importance of epigenomic reprogramming in cancer, with altered post-translational modifications of histones contributing to pathogenesis. However, the contributions of histone modifiers to breast cancer progression are unclear, and how these processes vary between molecular subtypes has yet to be adequately addressed. Here we report that genetic or pharmacological targeting of the epigenetic modifier Ezh2 dramatically hinders metastatic behaviour in both a mouse model of breast cancer and patient-derived xenografts reflective of the Luminal B subtype. We further define a subtype-specific molecular mechanism whereby EZH2 maintains H3K27me3-mediated repression of the *FOXC1* gene, thereby inactivating a FOXC1-driven, anti-invasive transcriptional program. We demonstrate that higher *FOXC1* is predictive of favourable outcome specifically in Luminal B breast cancer patients and establish the use of EZH2 methyltransferase inhibitors as a viable strategy to block metastasis in Luminal B breast cancer, where options for targeted therapy are limited.

[1] Goodman Cancer Research Centre, McGill University, Montréal, QC H3A 1A3, Canada. [2] Department of Biochemistry, McGill University, Montréal, QC H3G 1Y6, Canada. [3] McGill University, Génome Québec Innovation Centre, Montréal, QC H3A 0G1, Canada. [4] Department of Human Genetics, McGill University, Montréal, QC H3A 1A3, Canada. [5] Department of Surgery and Oncology, Jewish General Hospital, Montréal, QC H3T 1E2, Canada. [6] Departments of Medicine and Oncology, McGill University, Montréal, QC H3G 1Y6, Canada. Correspondence and requests for materials should be addressed to W.J.M. (email: william.muller@mcgill.ca)

Breast cancer is comprised of a variety of disease entities bearing diverse pathological features, prognostic outcomes and metastatic behaviours. In an effort to unravel the heterogeneity of this disease, global transcriptional profiling has led to the characterization of at least five different intrinsic molecular subtypes of breast cancer; HER2+, Luminal A, Luminal B, Normal-like and Basal-like breast cancer (BLBC)[1]. The Luminal B subtype has particularly poor survival outcome, partly due to the lack of viable therapeutic options. An important aspect of developing effective therapies consists of defining how distinct transcriptional networks within each subtype are uniquely modulated by genetic and epigenetic mechanisms. Trimethylation of lysine 27 on histone 3 (H3K27me3) by the methyltransferase Ezh2, as a part of the polycomb repressive complex 2 (PRC2), is an important mechanism of gene silencing. Interestingly, aberrant expression of Ezh2 has been widely observed in cancer, with reports of both oncogenic and tumour suppressive functions[2].

With respect to breast cancer, Ezh2 levels are observed to be elevated and increased expression has been associated with poor survival[3]. However, functional studies have failed to reach a consensus as to whether Ezh2 plays a causal role in driving disease or is merely a by-product of increased cellular proliferation[4]. The context-dependent nature of Ezh2 function has also been proposed to be determined by the cell of origin and/or early transformation events undertaken by a tumour cell[5]. Thus, the different developmental origins of intrinsic breast cancer subtypes[6] underscores the importance of evaluating the role of Ezh2 in each molecular subtype. Furthermore, while significantly higher H3K27me3 levels are observed in Luminal B compared to BLBC or HER2+, the functional relevance of this increased global histone methylation state is unclear[7].

Given that the behaviour of Ezh2 is context-dependent, in this study we investigated the role of Ezh2 specifically in Luminal B breast cancer. To this end, we employed a transgenic mouse model to examine the effects of Ezh2 ablation at each stage of tumorigenesis from early hyperplastic lesions to invasive metastatic disease. This work led to the identification of a Luminal B-specific anti-metastatic transcriptional program centred on the master transcriptional regulator FOXC1, which is silenced in these tumours in an Ezh2-dependent manner. Notably, pharmacological inhibitors targeting Ezh2 de-repressed FOXC1 and reactivated this anti-metastatic program in both murine and human preclinical models, resulting in a dramatic reduction in both the size and number of metastatic lesions. This is significant because the vast majority of breast cancer-associated morbidity and mortality are due to distant metastasis. Thus, our findings have important implications for the treatment of Luminal B breast cancer, where a paucity of options for targeted therapy has significantly hindered progress in improving patient outcomes.

## Results

**Ablation of Ezh2 impairs tumour onset and metastasis**. To explore the role of Ezh2 in Luminal B breast cancer, we employed a Polyomavirus Middle T (PyVmT)-driven model, in which the rapid development of tumours closely mimics human disease progression[8,9], and which has a transcriptional profile that clusters with that of the human Luminal B intrinsic subtype[10]. Since Ezh2 plays an important role in maintaining mammary luminal progenitor cells and is required for mammary alveologenesis[11,12], we utilized an inducible PyVmT transgene[13] to circumvent phenotypes arising from impaired mammary development (Tet-ON PyVmT). This model combines mouse mammary tumour virus promoter (MMTV)-driven expression of the reverse tetracycline-dependent transactivator (rtTA)[14] with a Tet

operator-controlled bicistronic transgene encoding the PyVmT oncogene and Cre recombinase. Hence, upon induction with doxycycline, coordinated PyVmT expression and deletion of conditional Ezh2 allele(s)[15] occur specifically in the mammary epithelium (Fig. 1a). Whereas cohorts of virgin females bearing the wild-type or one conditional allele of Ezh2 developed measurable mammary tumours after an average onset of 67 and 61 days, respectively, Ezh2-deficient mice displayed a significant delay in tumour onset to an average of 145 days (Fig. 1b). Ezh2 deletion and oncogene expression were confirmed by both immunoblot and immunohistochemistry analyses (Supplementary Figure 1A, B). Interestingly, Ezh2-deficient tumours assessed for H3K27me3 by immunofluorescence at endpoint exhibited undetectable levels of H3K27me3 in the tumour epithelium, indicating that Ezh1 was not able to compensate for the loss of Ezh2 histone methyltransferase activity (Fig. 1c, Supplementary Figure 1C, D).

An important, clinically relevant feature of the PyVmT model is its capacity to metastasize to the lungs with high efficiency. Since complete ablation of Ezh2 reduced tumour focality and proliferation (Supplementary Figure 1E–G), animals were killed at an endpoint defined by total tumour volume to confirm that metastatic phenotypes were exclusive of the reduced proliferative capacity of Ezh2-null tumours. Histological examination of lung tissue sections for the presence of metastatic lesions revealed that mice lacking Ezh2 exhibited a significant reduction in both the size and number of lesions (Fig. 1d, e). To further establish that the observed impact on metastasis was independent of differences in tumour burden and to gain insight into how loss of Ezh2 impairs the metastatic cascade, cells from freshly dissociated wild-type and Ezh2-null tumours were injected into the tail vein of athymic nude mice. Following 8 weeks of doxycycline induction, mice were killed and lungs were assessed. The results demonstrated that loss of Ezh2 significantly impairs the ability of cells to colonize the lung (Fig. 1f).

**Pharmacological H3K27me3 inhibition mimics genetic ablation**. In light of our genetic studies indicating that loss of Ezh2 impairs metastasis, we wished to explore the clinical relevance of these observations further and establish whether they were directly related to the methyltransferase activity of Ezh2. Several pharmacological inhibitors developed to target PRC2 activity have already shown clinical response in early phase trials in leukaemia[16,17]. To test the hypothesis that Ezh2 is a druggable anti-metastatic target, we treated immune-competent mouse models reflective of different clinical situations with the Ezh2 methyltransferase inhibitor GSK-126[18]. Previous characterization of the Tet-ON PyVmT model has shown that neoplastic lesions in the mammary gland can rapidly form following induction with doxycycline for 2 weeks, with full progression to invasive adenocarcinoma with metastasis to the lungs occurring by 6 weeks of doxycycline administration. Thus, to recapitulate a neoadjuvant therapy scenario following the detection of an aggressive early breast cancer, we induced Tet-ON PyVmT mice with doxycycline for 2 weeks to allow for palpable lesions to form, and then treated mice with either GSK-126 or vehicle for 4 weeks to determine the effect on the progression of these lesions (Fig. 2a). The efficacy of GSK-126-mediated inhibition was confirmed by immunohistochemical analysis of H3K27me3 levels (Supplementary Figure 2A). Consistent with the genetic ablation of Ezh2, we observed a dramatic decrease in the induction of mammary hyperplasias and early adenomas (Supplementary Figure 2B) and a complete block in the formation of lung metastases in GSK-126-treated cohorts (Fig. 2b). To further support the concept that the anti-metastatic effect of GSK-126 was due to its effect on the

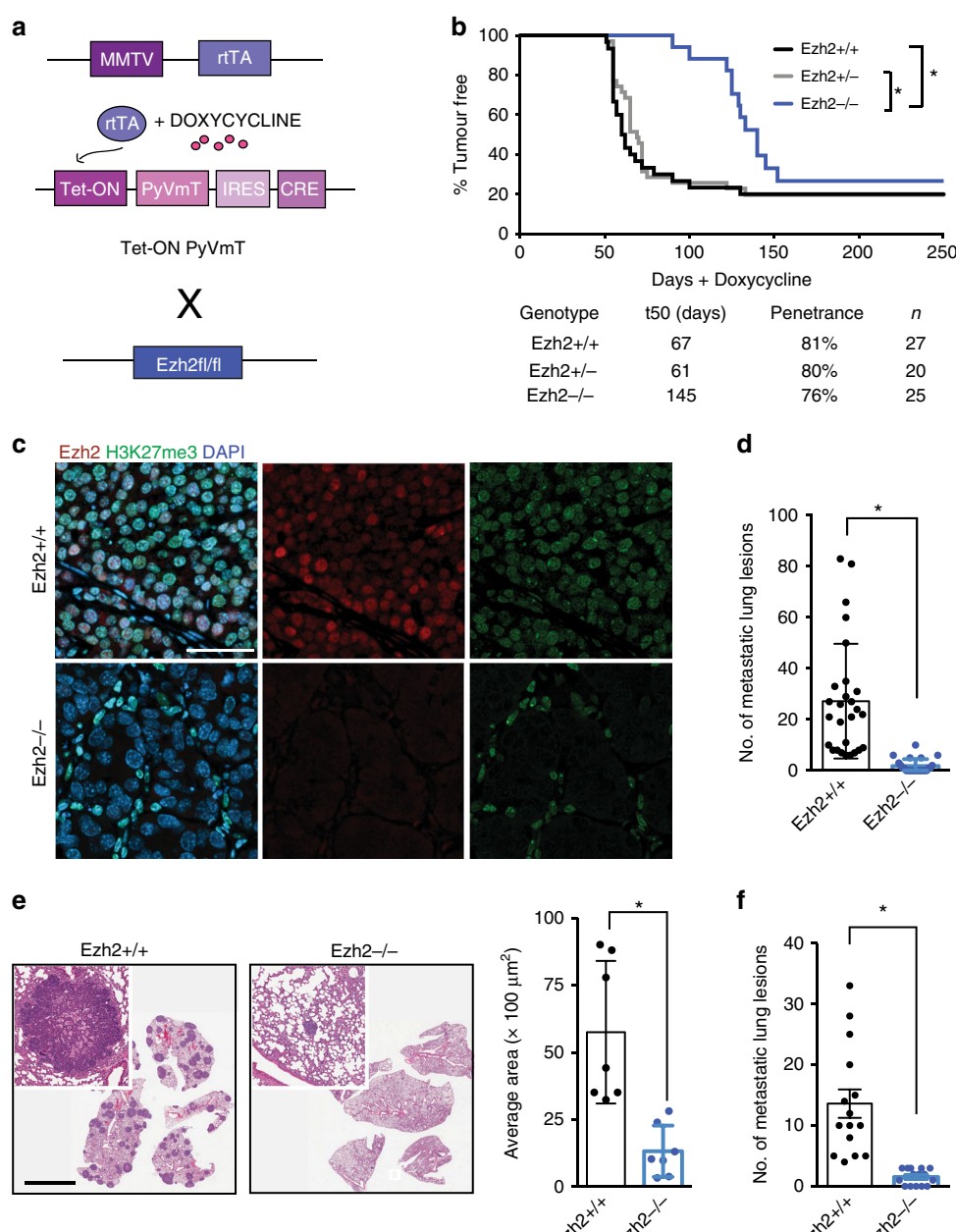

**Fig. 1** Loss of Ezh2 significantly alters breast cancer tumorigenicity. **a** Schematic of the transgenic mouse model. **b** Kaplan–Meier survival curve of mammary tumour onset in Tet-ON PyVmT mice with wild-type (Ezh2+/+, $n = 27$), heterozygous (Ezh2+/−, $n = 20$) or homozygous (Ezh2−/−, $n = 25$) Ezh2 conditional alleles. *$p < 0.05$, log rank test. **c** Immunofluorescence staining of endpoint Ezh2+/+ or Ezh2 −/− Tet-ON PyVmT tumours for Ezh2, H3K27me3 or the nuclear stain DAPI. Scale bar is 50 μm. **d** Average number of metastatic lung lesions from tumour burden endpoint Tet-ON PyVmT mice (Ezh2−/−, $n = 25$, Ezh2+/+, $n = 27$). *$p < 0.05$, Student's two tailed $t$-test. **e** Representative images of hematoxylin and eosin stained sections of paraffin-embedded lungs from Tet-ON PyVmT mice at tumour burden endpoint. Scale bar is 5 mm. Average area of lung lesions per genotype. *$p < 0.05$, Student's two tailed $t$-test. **f** Quantification of lung lesions following tail vein injections of freshly dissociated cells from Ezh2+/+ or Ezh2 −/− Tet-ON PyVmT tumours into athymic nude hosts. Mice were killed after 8 weeks and maintained on water supplemented with doxycycline for the duration of the experiment. $n = 15$ for each genotype. *$p < 0.05$, two tailed $t$-test

tumour cells, we examined the presence of spontaneous metastatic lung lesions in cohorts of wild-type and Ezh2-null Tet-ON PyVmT mice induced with doxycycline for 6 weeks. In line with our observations using GSK-126, metastatic lung lesions were detectable in all wild-type Tet-ON PyVmT mice, but not observed in any of the Ezh2-null Tet-ON PyVmT mice (Fig. 2b).

Given that Ezh2 has effects on both tumour initiation and metastasis in our model, we next determined whether GSK-126 administration can alter the metastatic potential of established

PyVmT tumour cells. To accomplish this, PyVmT cells were orthotopically implanted into the mammary fat pad of immune-competent hosts, and allowed to grow to a palpable size, at which point mice were treated with GSK-126 or vehicle (Fig. 2c). The effect of loss of H3K27me3 due to GSK-126 administration on tumour growth was not significant, although there was a small but significant difference in the tumour mass at the experimental endpoint (Supplementary Figure 2C,D). However, like genetic ablation of *Ezh2*, GSK-126 treatment resulted in a dramatic

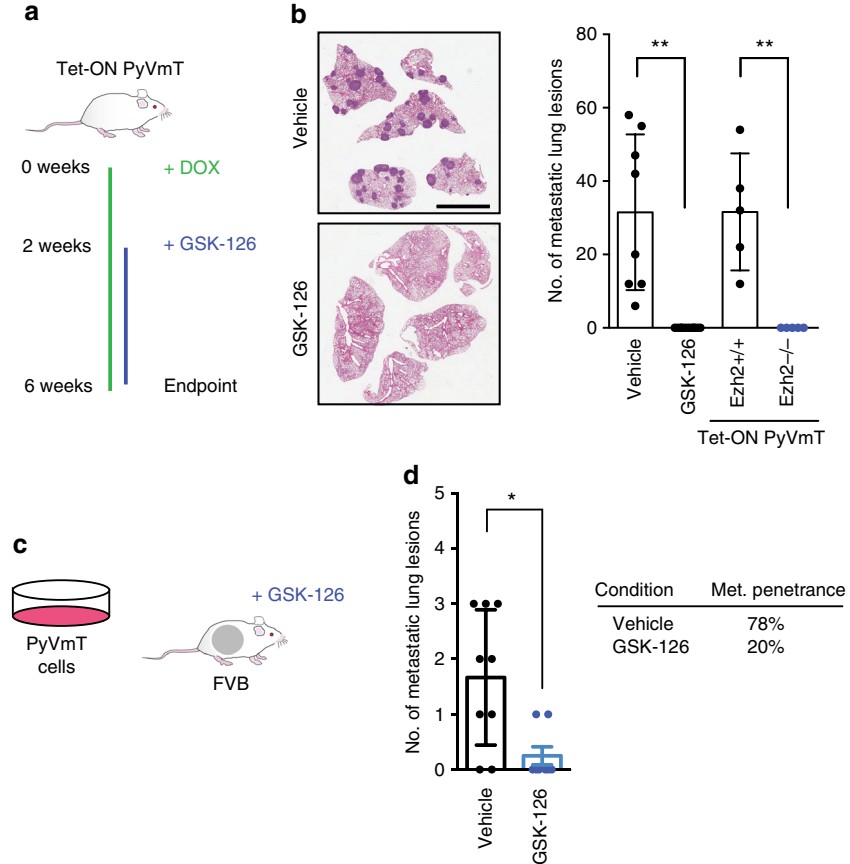

**Fig. 2** Pharmacological inhibition of Ezh2 methyltransferase inhibits metastasis in vivo. **a** Schematic of the preclinical model. **b** Representative images of hematoxylin and eosin stained paraffin-embedded lungs from mice treated with vehicle control (Captisol) or GSK-126 (150 mg/kg) via intraperitoneal injection, three times a week. Mice were killed after 6 weeks on water supplemented with doxycycline. Enumeration of metastatic lung lesions in drug treated cohorts as outlined in the schematic, or in Ezh2+/+ or Ezh2−/− Tet-ON PyVmT mice induced with doxycycline for 6 weeks. Scale bar is 5 mm. *$p$ < 0.05, two tailed $t$-test. **c** Schematic of orthotopically injected PyVmT cells in an immune-competent FVB strain host. GSK-126 or Captisol vehicle control treatment commenced when tumours reached 5×5 mm. **d** Enumeration of lung lesions from mice treated with GSK-126 or vehicle for 24 days. *$p$ < 0.05 **$p$ < 0.01, two tailed $t$-test. Met. penetrance, penetrance of metastatic lung lesions

reduction in metastatic burden (Fig. 2d). Considered together, these data suggest that the significant anti-metastatic effects of genetic or pharmacological targeting of Ezh2 in multiple models reflect a direct requirement for the catalytic activity of Ezh2 in facilitating dissemination of Luminal B breast cancer cells which is independent of the relatively minor effects on the growth of the primary tumour.

**Identification of a Foxc1-driven anti-metastatic cascade**. To evaluate the effect of loss of Ezh2 on the transcriptome of PyVmT tumours, gene expression profiling of Ezh2-proficient and Ezh2-deficient Tet-ON PyVmT endpoint tumours was conducted (Fig. 3a). Ingenuity pathway analysis (IPA) was used to reveal functional pathways and functions activated in the absence of Ezh2. The functional analysis uncovered a significant number of differentially regulated genes in Ezh2 knock-out tumours that were associated with cellular differentiation and axonal guidance (Supplementary Figure 3A).

Notably, Ezh2 has been reported to be necessary for epithelial to mesenchymal transformation (EMT), a process thought to be critical for metastatic disease progression[19]. However, functional enrichment analysis using IPA identified EMT as a biological pathway upregulated in Ezh2-null Tet-ON PyVmT tumours (Supplementary Figure 3A), suggesting that a block in EMT is

unlikely to explain the impaired metastatic capacity of these tumour cells. To independently validate the pathways identified by IPA, we also used Gene Set Enrichment Analysis (GSEA) to analyse our microarray data set. Similar to the IPA analysis, GSEA identified upregulation of pathways related to ECM organization and Axon Guidance in Ezh2−/− tumours (Supplementary Figure 3B).

We speculated that the metastatic impairment observed might be due to the activation of novel transcriptional programs in the absence of Ezh2-mediated repression. To explore this idea, we performed chromatin immunoprecipitation coupled with deep sequencing (ChIP-seq) of H3K27me3 on endpoint tumours. As anticipated, we observed a global decline in binding peak events across the genome including at the transcriptional start sites of genes in Ezh2-null tumours (Fig. 3b and Supplementary Figure 3C). Next, we intersected our transcriptomic and ChIP-seq data sets to identify genes directly regulated by Ezh2. Our cross-examination revealed that ~22% of differentially upregulated genes in Ezh2-null Tet-ON PyVmT tumours were H3K27me3 targets (Fig. 3c). Interestingly, these genes clustered into pathways associated with motility, neuronal identity and tissue morphogenesis (Supplementary Figure 3D).

Based on our hypothesis that Ezh2 silences anti-metastatic transcriptional programs in PyVmT tumours, we proceeded to identify potential regulatory factors involved in activating these

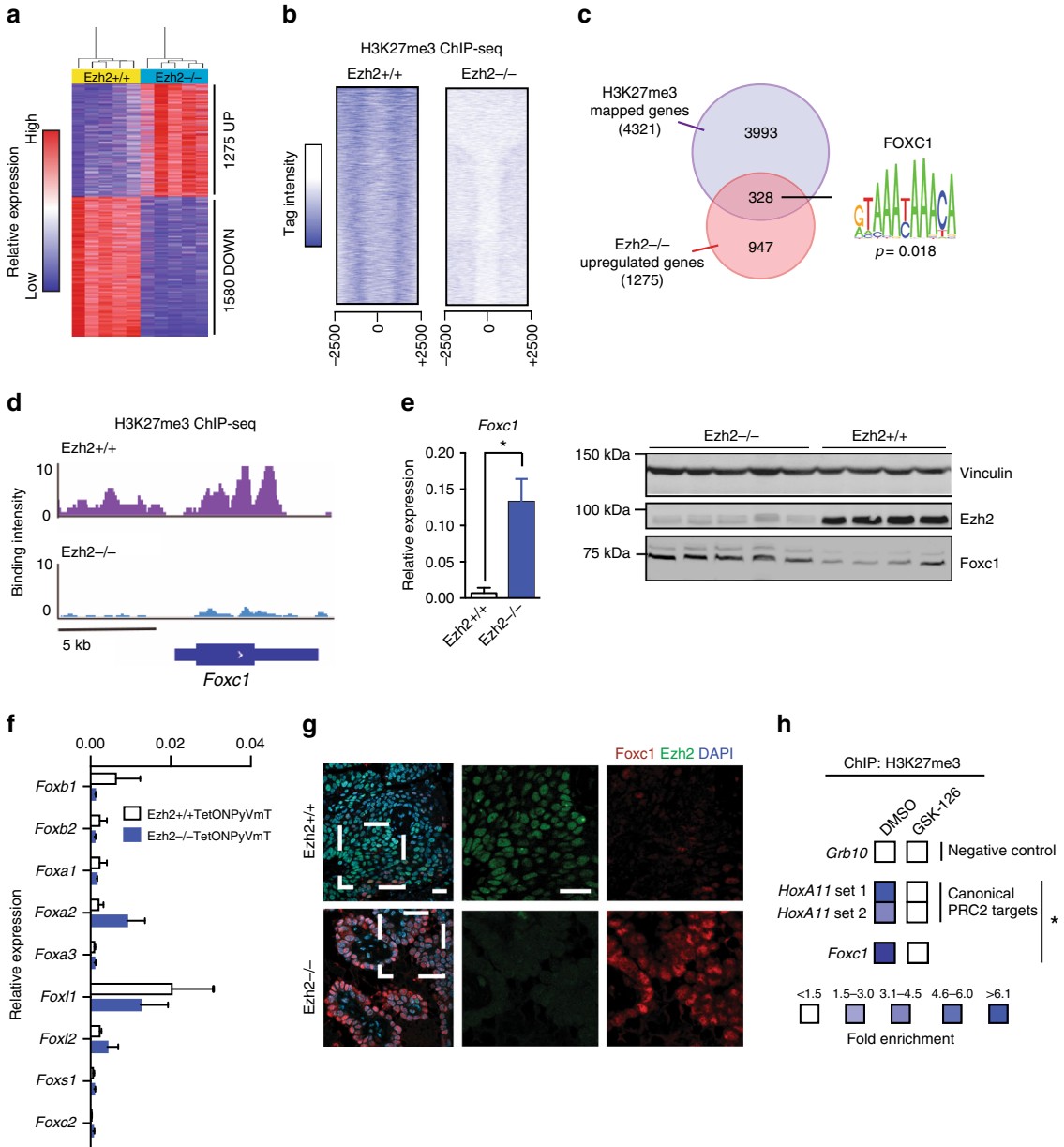

**Fig. 3** De-repression of Foxc1 following Ezh2 inhibition. **a** Hierarchical clustering of genes differentially expressed between Ezh2+/+ and Ezh2−/− Tet-ON PyVmt tumours (*n* = 5) using *p* < 0.05 and fold change > 1.5 cutoff. Red and blue indicate high and low expression of genes, respectively. **b** Heatmaps of signal intensity illustrating H3K27me3 ChIP-seq genomic mapping in a window of ±2.5 kb identified in Ezh2+/+ and Ezh2−/− Tet-ON PyVmt tumours. **c** Overlap between differentially upregulated genes in Ezh2−/− endpoint Tet-ON PyVmT tumours and genes identified by ChIP-seq to be targeted by H3K27me3. **d** Differential H3K27me3 levels in the upstream promoter region of *Foxc1* in Ezh2+/+ vs. Ezh2−/− endpoint Tet-On PyVmT tumours. Image from the IGV browser. **e** Left- Significant upregulation of *Foxc1* mRNA in Ezh2−/− Tet-ON PyVmT endpoint tumours compared to wild-type tumours. Right- Immunoblot of Ezh2+/+ or Ezh2−/− Tet-ON PyVmT endpoint tumours for Foxc1 and Ezh2 levels. Vinculin loading control. *\*p* < 0.05, two tailed *t*-test. **f** qRT-PCR screen of Forkhead box family members in tumours lacking Ezh2. **g** Immunofluorescence staining of Foxc1 and Ezh2 in endpoint Ezh2+/+ or Ezh2−/− Tet-ON PyVmT tumours. Scale bars are 50 μm. **h** Chromatin immunoprecipitation enrichment of H3K27me3 at canonical PRC2 targets and the *Foxc1* promoter in DMSO treated PyVmT cells which is lost in GSK-126 treated cells. *\*p* < 0.05, two tailed *t*-test

programs. To this end, we performed transcription factor motif enrichment analysis of the subset of genes with elevated expression in Ezh2-null tumours that also underwent loss of H3K27me3 from their promoter regions. Using several curated databases queried by the Enrichr web interface, or the motif search function of the HOMER computational suite of tools, we first demonstrated that PRC2 components were predicted to be upstream of the 328 overlapping genes in this set, confirming their status as PRC2 target genes (Supplementary Figure 3E).

These bioinformatic analyses also identified several potential transcriptional activators that could drive the expression of de-repressed PRC2 target genes in Ezh2-deficient tumours (Supplementary Figure 3F). Among transcription factors predicted to bind the promoters of these genes and regulate their expression, the candidate master regulatory factor Forkhead Box C1 (Foxc1) was of specific interest, because it was the only candidate factor that appeared in multiple analyses, and the only candidate significantly upregulated in our Ezh2−/− endpoint tumours

(Supplementary Figure 3F). By interrogation of our ChIP-Seq data to identify genes losing H3K27me3 in Ezh2−/− compared to Ezh2+/+ tumours, we confirmed that Foxc1 is directly targeted by Ezh2-mediated H3K27me3 (Fig. 3d). We further confirmed Foxc1 upregulation in the epithelial compartment of Ezh2-null tumours by qPCR and immunoblotting (Fig. 3e). Of note, tumours lacking Ezh2 did not have altered expression of other Forkhead box family members (Fig. 3f). Consistent with the notion that Foxc1 was a target of transcriptional silencing by Ezh2, immunofluorescence revealed that Ezh2 and Foxc1 did not co-localize in endpoint Tet-ON PyVmT tumours (Fig. 3g). As observed with the genetic loss of Ezh2, inhibition of Ezh2 activity with GSK-126 resulted in a specific increase in Foxc1 transcript levels (Supplementary Figure 3G) coinciding with loss of H3K27me3 in the promoter region of Foxc1 determined by ChIP-qPCR (Fig. 3h).

Given that Foxc1 is a transcription factor that can independently regulate its own target genes, we next assembled a set of 23 putative Foxc1 target genes (derived from the JASPAR database) that were both upregulated in the Ezh2−/− mice and enriched for H3K27me3 binding in the ChIP-Seq data set (Fig. 4a). Then, to understand the pathways downstream of the identified Foxc1 targets, we used Gene Ontology (GO) terms to organize Foxc1 targets found upregulated in the context of Ezh2 deletion (Fig. 4b). Several of these genes clustered with cellular processes related to neuronal function and ECM organization, and were reflective of the primary programs identified by IPA and GSEA in the broader Ezh2−/− upregulated gene set (Supplementary Figure 3,A,B). Interestingly, the overexpression of several of these targets has been reported to be anti-metastatic. The ECM matrix components Col4a6[20] and Col15a1[21] maintain basement membrane integrity to prevent tumour migration, and are lost prior to tumour invasion. Facilitators of axon guidance and cell adhesion Slfn5[22], Rgma[23,24] and Chl1[25] are also downregulated during breast cancer progression. To determine whether the effects of Foxc1 de-repression could be recapitulated pharmacologically, we validated the transcriptional upregulation of 13 Foxc1 target genes following treatment with GSK-126 (Fig. 4c). Next, we used ChIP-qPCR to validate that Foxc1 is indeed recruited to these targets upon global reduction of H3K27me3 in the absence of Ezh2 methyltransferase activity. Foxc1 occupancy in the promoter regions of a subset of putative Foxc1 targets (Rgma, Chl1, Slc9a9, Col15a1) was confirmed only in the presence of GSK-126 (Fig. 4d).

**Functional validation of Foxc1 anti-metastatic capacity.** Many of the validated downstream targets of Foxc1 exert their anti-metastatic effects through impairment of cellular motility and invasiveness. Thus, to test the contribution of Foxc1 to the metastatic cascade, we generated PyVmT cells stably expressing Foxc1 and assayed invasiveness. Interestingly, while ectopic expression of Foxc1 significantly impaired the ability of cells to invade across a basement membrane matrix (Fig. 5a, Supplementary Figure 4A), it did not alter cellular proliferation (Fig. 5b). In addition, the capacity of cells to colonize the lung was also dramatically diminished upon exogenous Foxc1 expression (Fig. 5c). Given these results, we sought to ascertain the extent of the contribution of Foxc1 to the metastatic defect induced by global reduction of H3K27me3 profiles. To this end, lentiviral-mediated RNA interference was used to knock-down expression of Foxc1 in PyVmT cells, which were then treated with GSK-126 or DMSO (vehicle). Expression of shRNAs targeting Foxc1 significantly blunted the de-repression of Foxc1 by Ezh2 inhibition (Fig. 5d, Supplementary Figure 4B). Remarkably, the loss of Foxc1 almost completely restored the invasiveness of the GSK-

126-treated cells to the level of control cells (Fig. 5d) indicating that transcriptional repression of Foxc1 is a critical molecular component of the pro-metastatic effect facilitated by H3K27me3-mediated silencing.

**Differential association of elevated FOXC1 across subtypes.** Having established that FOXC1 can drive an anti-metastatic program, we sought to confirm the relevance of this finding in a clinical context using both publicly available breast cancer patient data sets and human patient specimens.

To validate the relationship between EZH2 and FOXC1, we first quantified FOXC1 protein levels in 20 Luminal B patient tumour samples via immunofluorescence staining of breast tissue biopsies. We confirmed significant positive correlation with matched FOXC1 mRNA expression levels ($r = 0.57$, $p = 0.004$) (Fig. 6a and Supplementary Figure 5A). Importantly, we observed an inverse correlation between EZH2 and FOXC1 protein levels in this Luminal B cohort (Fig. 6b). This observed mutual exclusivity supported the mechanism of EZH2-mediated FOXC1 repression in Luminal B breast cancer.

To next address the prognostic value of FOXC1 levels across different subtypes, we interrogated a large breast cancer clinical data set available through KM plotter[26]. We observed that higher expression of FOXC1 was associated with increased relapse-free survival in Luminal B patients (Fig. 6c), but not in BLBC (Supplementary Figure 5B). Interestingly, there was a trend towards increased relapse-free survival in HER2+ patients with increased FOXC1 levels, and a significantly increased probability in Luminal A patients (Supplementary Figure 5,C,D). Taken together, these data suggested that FOXC1 plays a tumour suppressive role in non-BLBC subtypes and we hypothesized that the previously identified FOXC1 targets might facilitate this anti-metastatic program. To confirm this hypothesis, we interrogated the TCGA database using the previously validated FOXC1 13-gene signature (Fig. 4b). As expected, our analysis identified a positive correlation between FOXC1 transcript levels and the FOXC1 multi-gene signature in Luminal B patients (Fig. 6d) as well as the HER2+ and Luminal A subtypes, but not in the BLBC cohort (Supplementary Figure 5E–G). This observation for BLBC patients is consistent with previous findings, as FOXC1 has gained notoriety in breast cancer as a key marker for BLBC[27] dictating aggressiveness of the subtype through various mechanisms[28]. Hence, activation of an anti-metastatic gene signature by FOXC1 would not be expected in this subtype.

To establish the prognostic importance of the FOXC1 gene signature, we revisited KM plotter to analyse the effect of the FOXC1 signature levels across different breast cancer subtypes. Interestingly, higher expression of the FOXC1 signature conferred a significantly greater probability of relapse-free survival in only the Luminal B (Fig. 6e) and HER2+ breast cancer patients (Supplementary Figure 5H, I).

**EZH2-mediated metastasis suppression is Luminal B specific.** While FOXC1 may operate as an anti-metastatic factor in non-BLBC breast cancer subtypes, it was unclear if EZH2-mediated repression of an anti-metastatic program through suppression of FOXC1 was subtype-specific. To explore this, we first analysed EZH2 and FOXC1 transcript levels in a large cohort of breast cancer patient data from The Cancer Genome Atlas (TCGA) database. Notably, we observed that higher levels of EZH2 mRNA reads coincided with lower levels of FOXC1 read counts only in the Luminal B and HER2+ subtypes (Fig. 7a). Given that we observed this inverse association in both Luminal B and HER2+ patient samples, next we examined whether EZH2-mediated H3K27me3 repression of FOXC1 was unique to the Luminal B

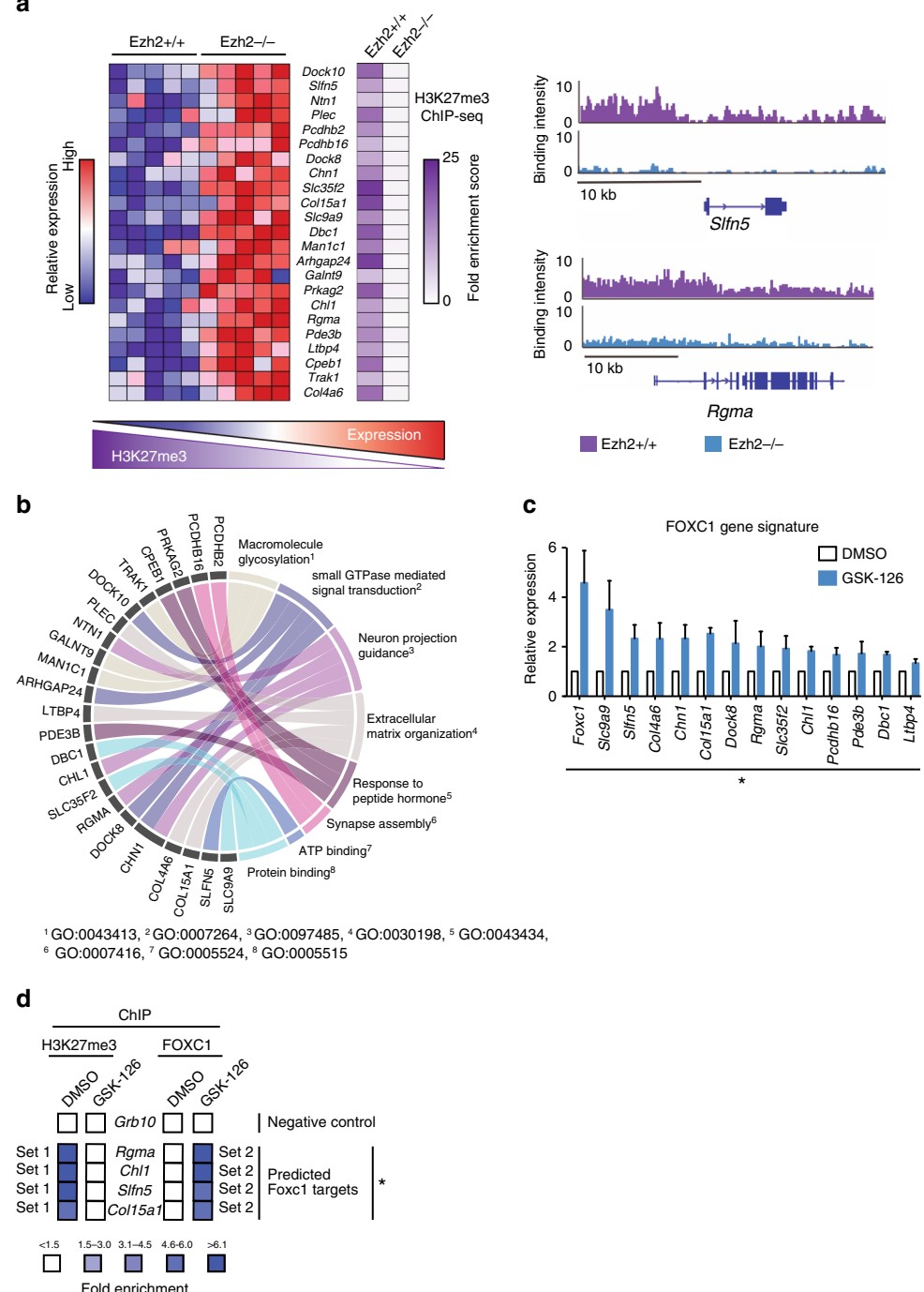

**Fig. 4** Identification of anti-metastatic targets repressed by Ezh2. **a** Increased expression of Foxc1 predicted target genes coincides with reduced H3K27me3 enrichment status in Ezh2−/− versus Ezh2+/+ tumours. Right panel shows a comparison of H3K27me3 binding intensity between Ezh2+/+ and Ezh2−/− tumours for two known targets of FOXC1 activation, taken from the IGV browser. **b** Chord diagram of upregulated Foxc1 targets and their associated GO Biological Terms. The ChIP experiment is reflective of three different PyVmT cell lines. **c** Quantitative RT-PCR analysis of Foxc1 targets in PyVmT cells treated with GSK-126 (2 μM for 72 h) or DMSO (vehicle). Data is an average of experiments performed in three different PyVmT cell lines. *$p$ < 0.05 (**d**) Chromatin immunoprecipitation of H3K27me3 or Foxc1 in the presence or absence of Ezh2-mediated H3K27me3 profiles. Three different PyVmT cell lines were treated with or without GSK-126 (2 μM, 72 h) or DMSO. Specific primers to detect binding by H3K27me3 and Foxc1 were designed. *$p$ < 0.05, two tailed $t$-tests

subtype. First, we treated a panel of human breast cancer cell lines representing four different molecular subtypes with GSK-126. Subsequent analysis of transcript levels demonstrated that only the Luminal B cell lines showed significant upregulation of *FOXC1* upon reduction of global H3K27me3 levels

(Supplementary Figure 6A). To confirm the subtype-specific anti-metastatic capacity of FOXC1, we exogenously expressed FOXC1 in two different Luminal B human breast cancer cell lines and assayed invasiveness via a Boyden chamber assay. As with PyVmT cells, we found that FOXC1 significantly suppressed

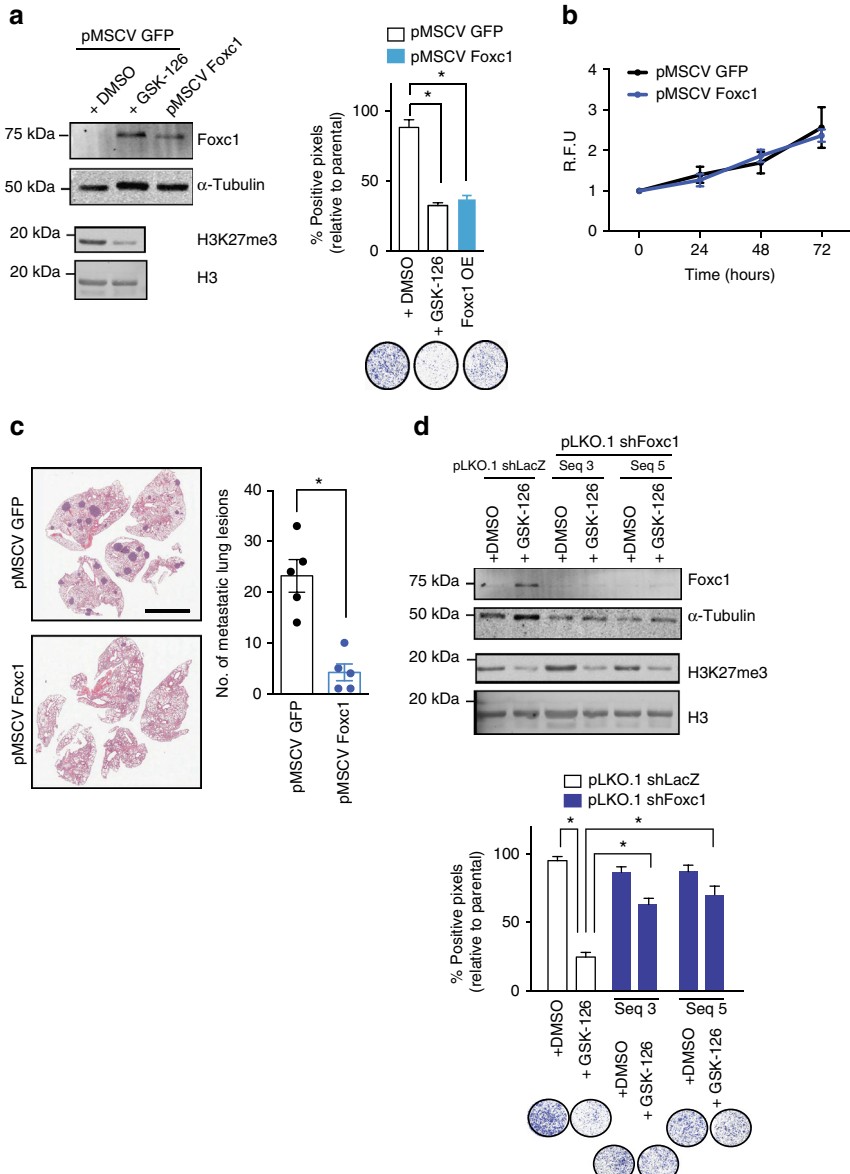

**Fig. 5** Ectopic expression of Foxc1 alters invasion but not proliferation. **a** Transwell invasion assay of PyVmT cells expressing exogenous Foxc1 or GFP. Experiments were performed three times, and results are displayed relative to untreated parental cell. *$p < 0.05$, Student's two tailed $t$-tests. **b** CyQUANT assay measuring cellular proliferation of PyVmT cells expressing exogenous GFP or Foxc1. R.F.U relative fluorescence units. Data were normalized to the fluorescence values at time = 0. **c** Enumeration of lung lesions following the injection of PyVmT cells expressing exogenous Foxc1 or GFP in the tail vein of athymic nude mice. *$p < 0.05$, Student's two tailed $t$-tests. Scale bar is 5 mm. **d** Left: Representative immunoblot showing Foxc1 expression in cells stably transduced with a control shRNA (shLacZ) or two independent shRNAs targeting *Foxc1* and treated with DMSO or GSK-126 (2 μM for 72 h). α-Tubulin was used as a loading control. Right: Transwell invasion assay of PyVmT cells infected with shRNA targeting LacZ (control) or two different sequences targeting Foxc1. Cells were pre-treated with DMSO or GSK-126 (2 μM) for 72 h and assayed for their capacity to invade through Matrigel. All assays were performed in triplicate. *$p < 0.05$, Student's two tailed $t$-tests

invasion in these human Luminal B cell lines (Supplementary Figure 6B). Conversely, we also confirmed that FOXC1 depletion could rescue reduced invasiveness mediated by administration of GSK-126 in two separate human Luminal B breast cancer cell lines in vitro (Supplementary Figure 6C).

To interrogate further correlations between FOXC1, EZH2 and downstream FOXC1 targets at the protein level in human breast cancer, we made use of recently published proteomic analysis of a subset of TCGA breast cancer patients[29]. In line with published observations, we observed that FOXC1 levels were elevated in BLBC patients compared to all other subtypes (Supplementary Figure 6D, E). In agreement with our immunofluorescence and

RNA-level analyses, we observed anti-correlation between EZH2 and FOXC1 protein levels specifically among Luminal B patients. Furthermore, protein levels corresponding to FOXC1 target genes were lower in Luminal B patients with high EZH2 and low FOXC1 expression, supporting our hypothesis that a FOXC1-driven transcriptional program is suppressed by EZH2 specifically in Luminal B breast cancer.

Finally, to confirm the anti-metastatic effect of EZH2 inhibition in distinct molecular subtypes of human breast cancer in vivo, we orthotopically transplanted patient-derived xenografts (PDX) established from Luminal B (HCI003, GCRC 1986)[30] or HER2+ (GCRC1991, GCRC 2080) samples into the mammary

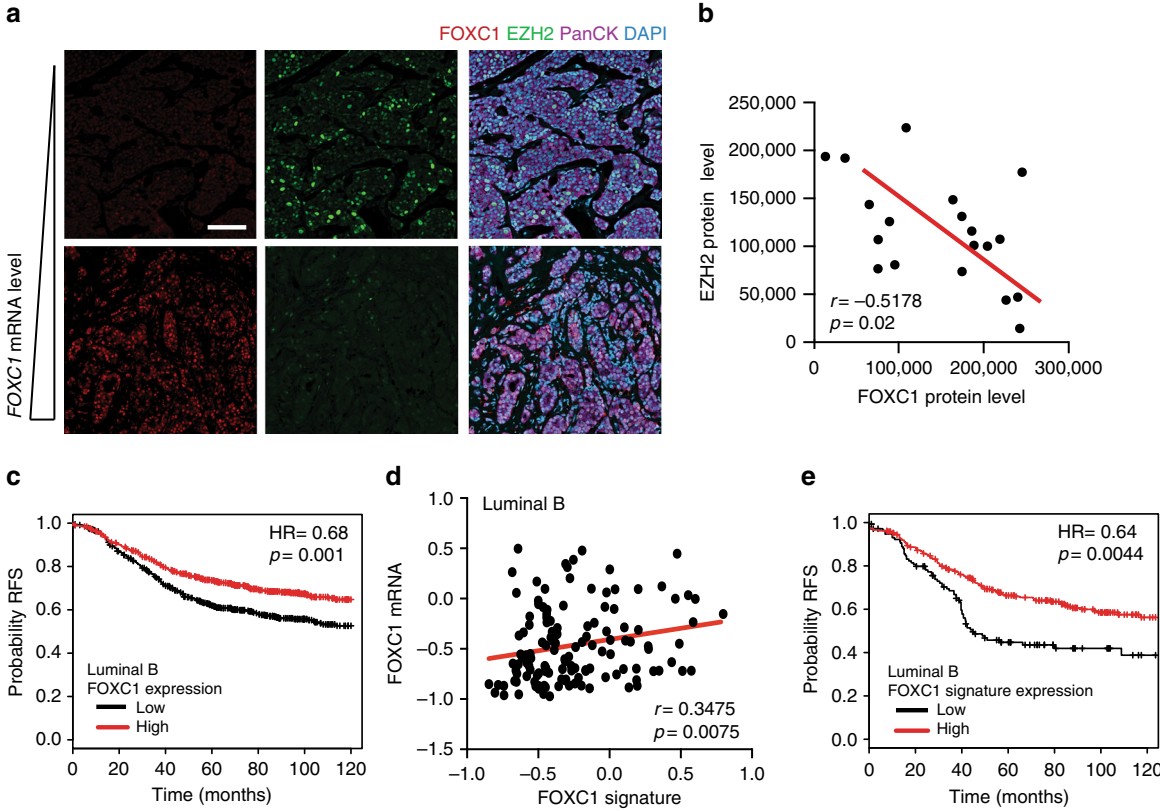

**Fig. 6** Breast Cancer subtype specificity of the EZH2/FOXC1 metastatic axis. **a** Representative images of formalin fixed paraffin-embedded tissue samples from human Luminal B tumours stained for FOXC1 and EZH2. Scale bar is 100 μm. **b** Significant negative correlation between FOXC1 transcript levels and EZH2 protein levels (Total intensity) quantified by immunofluorescence in human Luminal B patient samples ($r = -0.52$, $p = 0.02$). Eight different fields taken with a ×20 objective were quantified per sample. **c** Kaplan–Meier survival curve of relapse-free survival (RFS) of 1142 luminal B tumours with high or low FOXC1 (log rank $p = 0.001$). **d** Correlation between *FOXC1* mRNA levels and the FOXC1 Signature in a cohort of 385 Luminal B patients from publicly available TCGA Breast Cancer data set. **e** Kaplan–Meier analysis of relapse-free survival of 1142 Luminal B tumours with high or low FOXC1 gene signature expression (log rank $p = 0.0044$). RFS relapse-free survival

fat pads of immune compromised hosts and treated mice with either GSK-126 or vehicle for 5 weeks. Spontaneous lung metastases in hosts transplanted with the Luminal B PDXs were identified via a combination of human-specific pan cytokeratin immunohistochemical staining and enlarged nuclear morphology (Fig. 7b), and in hosts transplanted with HER2+ PDXs by immunohistochemical staining to detect HER2 (Supplementary Figure 7A). Strikingly, in the HCI003 Luminal B PDX, we observed only a 40% penetrance of lung metastases in the GSK-126 treated cohort ($n = 2/5$), compared to 100% in the vehicle control group ($n = 5/5$). Among the mice that did get lung metastases, the number of lung lesions per mouse was significantly decreased in the GSK-126 treated cohort compared to the vehicle control in the Luminal B, but not the HER2+ PDX hosts (Fig. 7c). Of the lung metastases that did appear in GSK-126 treated mice transplanted with Luminal B HCI003 PDX, the vast majority were micrometastases (<10 cells), with only 7% (1/14 total enumerated lesions) defined as overt metastases (Supplementary Figure 7B). However, in vehicle treated hosts, 30% (33/109 total enumerated lesions) of metastatic lesions were identified as overt metastases. In the context of the two independent HER2+ PDX models, the penetrance or number of lung metastases between treatment groups (Fig. 7c) and the relative proportions of micro vs overt metastases (Supplementary Figure 7B) were unaffected by GSK-126 treatment. Interestingly, while we observed a pronounced effect of GSK-126 on lung metastasis in both Luminal B PDXs, we did not observe any significant effect

on tumour growth (Supplementary Figure 7C), though a significant effect on tumour growth was observed in the HER2 + PDX GCRC1991 (Supplementary Figure 7D). The Luminal B PDX findings are in line with our results generated from preclinical studies in transgenic mouse models and indicate that the loss of global H3K27me3 levels directly perturbs the metastatic cascade rather than primary tumour growth in the Luminal B subtype. We also observed that in vivo global depletion of H3K27me3 levels did not significantly affect tumour growth in two separate TNBC PDXs (Supplementary Figure 7E), nor did it decrease metastatic burden. TNBC PDX GCRC 1735 was not metastatic and thus could not be assessed, but we observed that GSK-126 did not influence spontaneous lung metastasis in the TNBC PDX HCI010 (Supplementary Figure 7F).

Finally, to confirm that *FOXC1* levels were indeed elevated in Luminal B but not HER2+ PDXs treated with GSK-126, we performed qRT-PCR on breast tumour samples at endpoint and confirmed a significant upregulation of *FOXC1* transcript levels in GSK-126 treated HCI003 Luminal B but not 1991 HER2+ PDXs (Fig. 7d). Additionally, we observed the significant upregulation of four genes previously confirmed to be FOXC1 targets in Luminal B PDXs treated with the Ezh2 inhibitor (Fig. 7e).

Taken together, these data provide compelling evidence that pharmacological intervention targeting the EZH2-dependent repression of a FOXC1-driven transcriptional program can hinder the metastatic properties of human Luminal B breast cancer (Fig. 8).

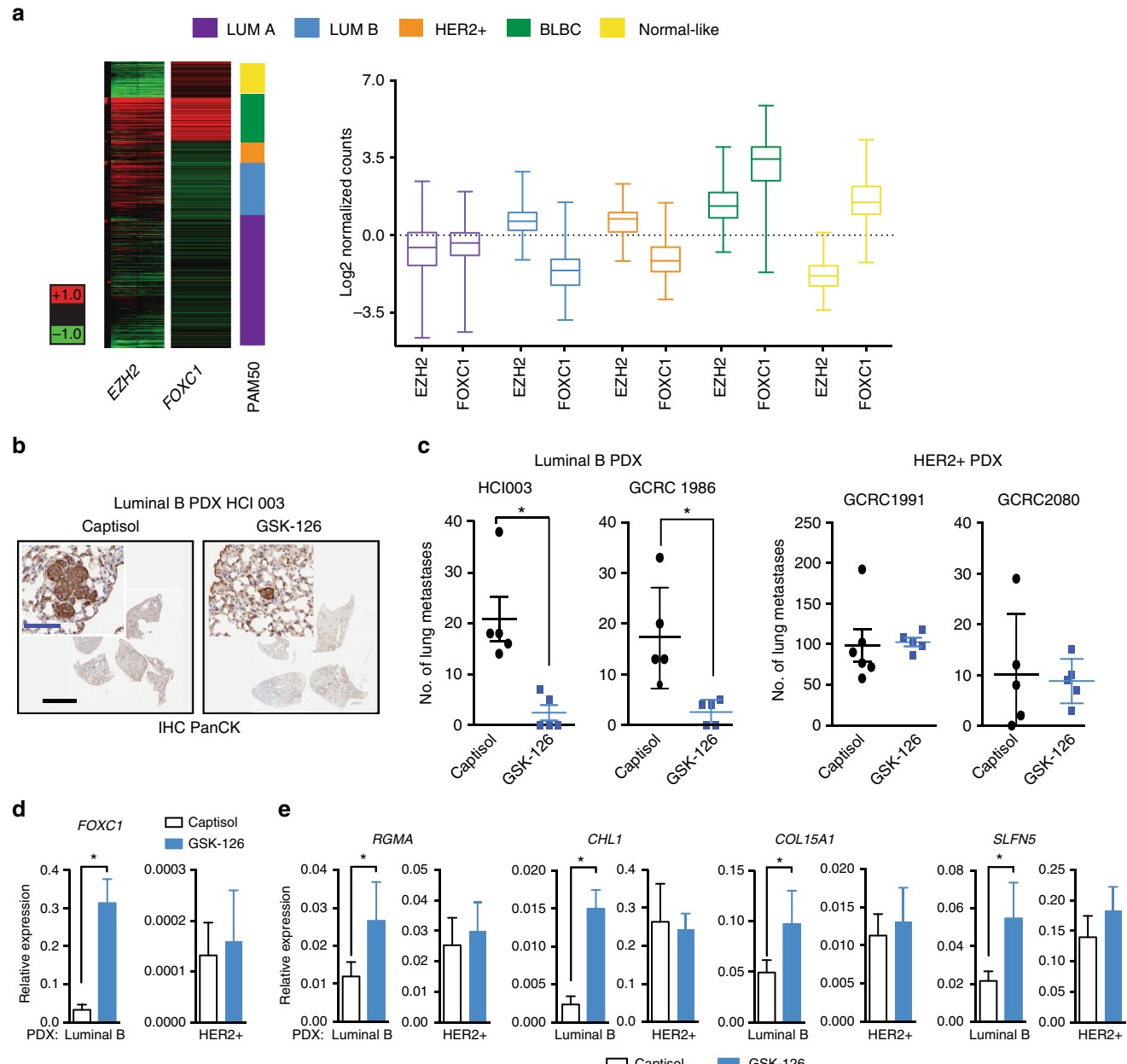

**Fig. 7** Inhibition of Ezh2 methyltransferase activity reduces the metastatic capacity of Luminal B but not HER2 breast cancer. **a** *FOXC1* or *EZH2* transcript levels across different PAM50 subtypes of breast cancer represented in a heat map. Turkey box plots showing median EZH2 or FOXC1 levels across different breast cancer subtypes. Data is from the TCGA Breast Cancer data set, publicly accessible via the XENA UCSC Cancer Genome Browser. Dotted Red outline denotes Luminal B subset. Normal-like (yellow), $n = 119$, EZH2 s.d $= 0.72$, mean $= 1.54$, FOXC1 s.d $= 1$, mean $= -1.64$. BLBC (green), $n = 142$, EZH2 s.d $= 0.81$, mean $= 1.3$, FOXC1 s.d $= 1,27$, mean $= 3.18$. HER2 (orange), $n = 67$, EZH2 s.d $= 0.73$, mean $= 0.63$, FOXC1 s.d $= 0.84$, mean $= -1.13$. Luminal A (purple), $n = 434$, EZH2 s.d $= 1.08$, mean $= -0.64$, FOXC1 s.d $= 0.83$, mean $= -0.44$. Luminal B (blue) $n = 194$, EZH2 s.d $= 0.65$, mean $= 0.60$, FOXC1 s.d $= 0.89$, mean $= -1.64$. Whiskers for the plots signifiy s.d, while the bar denotes the mean for each subtype. **b** Representative images of paraffin-embedded sections of lung stained with pan cytokeratin (pan CK). Lung lesions were identified by pan CK positivity and enlarged nuclei. Top scale bar is 200 μm. Bottom scale bar is 5 mm. **c** Quantification of lung lesions following the implantation of Luminal B PDX (HCI003) or HER2+ PDX (GCRC1991) into the mammary fat pad, followed by treatment with GSK-126 or vehicle (Captisol). $N = 5$ per condition. $*p < 0.05$, 1 tailed $t$-test. **d** qRT-PCR analysis of *FOXC1* transcript levels in endpoint Luminal B ($n = 4$ per condition) or HER2+ ($n = 4$ per condition) PDX tumours treated with GSK-126 or vehicle long term in vivo. $* p < 0.05$, 1 tailed $t$-test. **e** qRT-PCR analysis of 4 FOXC1 targets in in endpoint Luminal B ($n = 4$ per condition) or HER2+ ($n = 4$ per condition) PDX tumours treated with GSK-126 or vehicle long term in vivo. $*p < 0.05$, 2 tailed $t$-test

## Discussion

The highly proliferative, metastatic and often treatment-refractory nature of Luminal B breast cancer underlies its poor prognosis. However, limited understanding of genetic and epigenetic events driving this subtype hinders the advancement of successful therapies. In this study, we identify a mechanism through which EZH2 mediates repression of *FOXC1* to promote metastasis in Luminal B breast cancer. Perhaps most significantly, we demonstrate that pharmacological intervention to inhibit EZH2 in an immune-competent preclinical mouse model of Luminal B breast cancer completely prevents distal metastasis (Fig. 2). We also utilized PDX models, arguably the most relevant

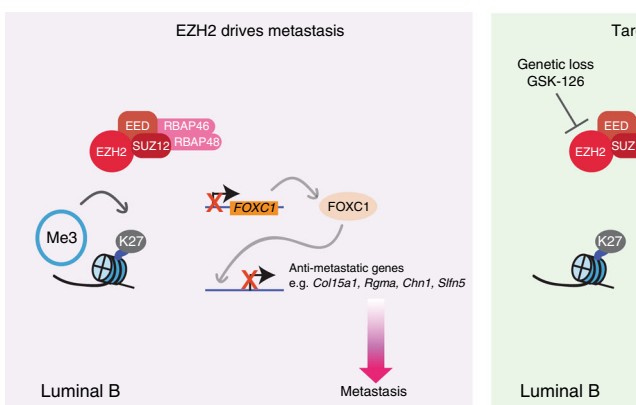
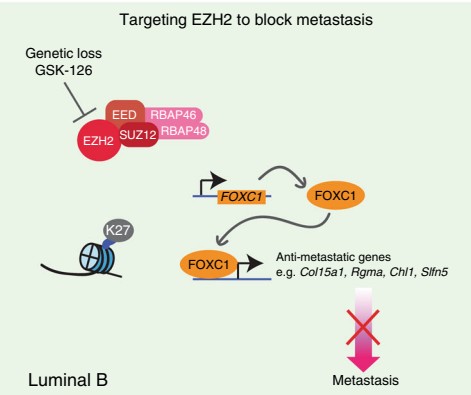

**Fig. 8** Proposed model highlighting the benefit of targeting Ezh2 in Luminal B breast cancer. Schematic illustrates Luminal B-specific PRC2-mediated suppression of a FOXC1-dependent anti-metastatic transcriptional program, and establishment of *FOXC1* expression following ablation of EZH2 or pharmacological inhibition of EZH2 methyltransferase activity leading to FOXC1-dependant expression of anti-metastatic genes

existing human preclinical models, to illustrate the significant effect of EZH2 methyltransferase inhibition on spontaneous metastasis in Luminal B but not HER2+ breast cancer (Fig. 7). These findings raise the possibility that EZH2 methyltransferase inhibitors may be used clinically to suppress invasion and metastasis in Luminal B breast cancer patients. While PRC2 function has predominantly been framed in the context of its histone methyltransferase activity, some studies have demonstrated the ability of PRC2 to methylate non-histone protein targets or for Ezh2 to modulate transcription independently of PRC2 in a cancer context[31–33], including in a transgenic model of HER2+ breast cancer where exogenous expression of Ezh2 enhanced the capacity of tumour-initiating cells in a PRC2-independent fashion[34]. While we cannot rule out potential roles for non-canonical Ezh2 functions in promoting mammary tumorigenesis in the models used here, we show that metastasis-suppressive functions of Ezh2 in Luminal B models involve its methyltransferase activity, while inhibition of endogenous EZH2 activity in ErbB2+ models had no effect on metastasis.

Our interest in pursuing the striking metastatic phenotype observed upon *Ezh2* ablation was driven by clinical knowledge that patient mortality is predominantly a consequence of distal metastasis. Interestingly, while our approach of using in vivo transgenic models allowed us to test progression of the metastatic cascade in a biologically relevant way, it also uncovered novel insights into the role of Ezh2 in tumour onset and progression. We observed that deletion of *Ezh2* in a PyVmT-driven model significantly delayed tumour onset, thus indicating that Ezh2 plays an important role in breast cancer progression. This directly contrasted with findings from recent studies that have utilized other transgenic mouse models to dissect the role of Ezh2 in breast cancer. Woo and colleagues employed a p53 heterozygous *Brca*-deficient mouse model of BLBC and speculated that the absence of any phenotype was due to compensating methyl-transferase activity by Ezh1[35]. This was not the case in our transgenic model, as evidenced by global depletion of H3K27me3 in the tumour epithelium (Supplementary Figure 1C,D). Wassef et al. utilized an activated Notch model of mammary tumorigenesis[4] and observed that ablation of *Ezh2* increased the penetrance of mammary tumours following multiple rounds of pregnancy. While it is unclear which intrinsic subtype is represented by the activated Notch model, we speculate that the difference observed could be due to differences in the cell of origin for each model. Ultimately, these conflicting data highlight that the notoriously context-specific nature of PRC2 function extends to different intrinsic subtypes of breast cancer. Thus, sweeping generalizations where breast cancer is treated as a single entity

with respect to the roles of PRC2 underestimate the biological heterogeneity of this disease.

Like EZH2, we found that FOXC1 functions in a context-specific manner in breast cancer. In general, it appears that the output of FOXC1 transcriptional regulation in non-BLBC subtypes of breast cancer differs significantly from that in BLBC. *FOXC1* is an established marker of EMT and poor outcome in BLBC[36], where it also suppresses ERα expression and may thereby contribute to reduced responsiveness to endocrine therapies such as tamoxifen[37,38]. In contrast, our functional studies demonstrate that higher *FOXC1* levels significantly correlate with good prognosis in non-BLBC subtypes of breast cancer. Furthermore, elevated expression of FOXC1 is detrimental to the metastatic cascade specifically within the Luminal B subtype. While further roles for FOXC1 in Luminal B breast cancers may include modulation of hormone receptor status and therapeutic response, our observations strongly suggest that any such pro-tumour effects of FOXC1 expression in Luminal B patients, if they occur, are overridden by the anti-metastatic function of FOXC1, leading to an overall pro-survival effect in Luminal B patients expressing higher FOXC1 levels (Refer to figures).

In addition to our observation that high *FOXC1* is a prognostic marker for positive outcome in Luminal A, HER2+ and Luminal B subtypes, we found that higher collective expression of a set of FOXC1 targets also predicted favourable prognosis in these patients (Fig. 6e and Supplementary Figure 5H). This suggests that the downstream targets of FOXC1 in these cancers are tumour suppressive, which agrees with multiple functional studies investigating these genes[20–25]. Typically, FOXC1 has been implicated in developmental processes such as blood vessel maturation and lymphatic sprouting[39]. Interestingly, characterization of FOXC1 targets in the Luminal B context revealed that they were associated with neuronal development. Given the importance of H3K27me3-dependent silencing during cell lineage decisions and differentiation[40], it is possible that the absence of EZH2 may have allowed for induction of an inappropriate differentiation program that incidentally also had anti-metastatic capacities. However, our findings suggest that the role of EZH2 in suppressing this program is subtype-specific, with inhibition of EZH2-reactivating FOXC1 and downstream targets only in Luminal B cell lines and PDX models (Supplementary Figures 6 and 7, Fig. 7). This could reflect the occurrence of different mechanisms of silencing FOXC1 across breast cancer subtypes. Interestingly, CpG islands near the *FOXC1* locus can be hypermethylated in early breast cancers[41]. DNA methylation and H3K27me3 are two important epigenetic silencing mechanisms that have been closely associated with each other[42], yet the precise

relationships are complex, with DNA methylation patterns potentially affecting the affinity of PRC2 binding and driving the redistribution of H3K27me3[43] or preventing H3K27me3 deposition locally and at a megabase scale[44]. Thus, it is conceivable that DNA methylation, H3K27 methylation, or potentially other mechanisms of silencing *FOXC1* may be adopted during tumour evolution in a subtype-specific manner. Furthermore, since loss of an epigenetic repressive mark does not necessarily guarantee expression of a given locus, a transcription factor milieu intrinsic to the Luminal B subtype might promote the active transcription of FOXC1 in the absence of repressive epigenetic marks.

In summary, this work establishes an oncogenic function for EZH2 in the development of Luminal B breast cancer and particularly in the metastatic phase of the disease. We have further established the potential of using FOXC1 as a prognostic biomarker in specific breast cancer subtypes and identified a pharmacological means to suppress metastatic progression in a subtype of breast cancer for which there is unmet clinical need.

## Methods

**Animals**. 8–12-week-old WT, Ezh2+/− or Ezh2−/− Tet-ON PyVmT female mice on a pure FVB background were induced with water supplemented with Doxycycline (200 mg/ml, Wisent). Both the Tet-ON PyVmT model[13] and Ezh2 conditional model[15] have been described previously. Mammary tumour formation was monitored by twice weekly palpation and tumour growth was measured by twice weekly caliper measurements. All mouse manipulations were performed in accordance with the McGill Facility Animal Care Committee and the Canadian Council on Animal Care.

**In vivo drug studies**. GSK-126 (Merca Chem) was reconstituted in 20% Captisol (Medchem Express), and brought to a pH of 4.5 with 10 M potassium hydroxide, to create a working stock of 150 mg/ml. Mice were weighed weekly, and administered 300 mg/kg of GSK-126 or an equal volume of vehicle via intraperitoneal injection, three times a week. Tumour growth was assessed in a blinded fashion, with twice weekly caliper measurements. Treatment groups were distributed, such that each cage had at least one mouse receiving each of the treatment conditions.

**Orthotopic transplantation of PyVmT cells**. For orthotopic transplantation, 500,000 cells from established PyVmT cell lines were suspended in 30 μl PBS and injected unilaterally into the inguinal fat pads of 8–12-week-old immunecompetent FVB mice.

**Tail vein injections**. For tail vein experiments, 250,000 cells were suspended in 100 μl PBS and injected in the tail vein of athymic nude mice. Mice injected with cells from freshly dissociated wild-type or Ezh2-null Tet-ON PyVmT tumours were maintained on water supplemented with doxycycline (200 mg/ml, Wisent) for the duration of the experiment (8 weeks). Mice injected with PyVmT cells expressing GFP or Foxc1 were killed after 4 weeks.

**ShRNA silencing and transfection of plasmids**. Exogenous Foxc1 expression in mouse cells was performed using the blasticidin selectable retroviral vector pMSCV-GFP (Gift from Dr. Peter Siegel). Foxc1 cDNA (MR208772, OriGene Gene Tech) was cloned into the vector. pMSCV-Foxc1 or pMSCV-GFP was transfected into Phoenix (obtained from the ATCC) cells using Lipofectamine 3000 (Invitrogen). Supernatants were collected 48 and 72 h post-transfection, passed through a 0.45 μm nitrocellulose filter and applied to target cells with polybrene (5 μg/ml, Sigma). Cells were re-infected the next day and selected with blasticidin for 72 h (10 μg/ml, Invitrogen). Five days after infection, cells were transferred to 10 cm dishes and maintained for experimental purposes. Foxc1 targeted knock-down in murine cells was performed using the Sigma MISSION pLKO.1 mouse Foxc1 shRNAs (clone ids TRCN0000085452 and TRCN0000085449) or LacZ shRNA (MFC07785395). Each shRNA vector was co-transfected into HEK293T cells with the lentivirus packaging plasmids PLP1, PLP2, and PLP-VSVG (Invitrogen) using Lipofectamine 3000 (Invitrogen). Supernatants were collected 48 and 72 h post-transfection, passed through a 0.45 μm nitrocellulose filter and applied on target cells with polybrene (5 μg/ml, Sigma). Cells were re-infected the next day and selected with puromycin for 48 h (1 μg/ml, Sigma). After 48 h post-infection, cells were transferred to 10 cm dishes and maintained for experimental purposes. In human cell lines, FOXC1 knock-down was performed using Dharmacon siGENOME siRNA (D-009318-02, D-009318-03) or a nontargeting siRNA pool (D-001206-13-05) using Lipofectamine 3000 (Invitrogen) and utilized for assays 72 h post-transfection. For the exogenous expression of FOXC1 cDNA (clone id X0042, GeneCopoeia) or GFP in the pReciever Lv-105

cDNA vector, human breast cancer cell lines were transfected with the appropriate vector using Lipofectamine 3000 (Invitrogen) and harvested for assays 24 h post-transfection.

**Cell culture**. Primary mouse tumour cell lines were established from PyVmT tumours by physical dissociation with a McIlwain Tissue Chopper, followed by incubation with DMEM supplemented with 2 mg/ml each of collagenase B/ Dispase II (Roche) for 1 h in at 37C. Primary mouse cells were maintained in Complete Media (DMEM, 10% FBS, 5 ng/ml EGF, 1 μg/ml Hydrocortisone, 5 μg/ml Insulin, 35 μg/ml Bovine Pituitary Extract. All components were from Wisent, except the Bovine Pituitary Extract, which was from Hammond Cell Tech). All human cell lines were purchased from ATCC and maintained in DMEM or RPMI supplemented with 10% FBS. Cell lines were routinely tested for mycoplasma using a commercially available kit (MycoAlert, Lonza).

**In vitro proliferation assay**. The CyQUANT proliferation assay (ThermoFischer Scientific) was performed in accordance with the manufacturer's protocols using 2500 cells per well in 96-well optical-bottom plates (Nunc). Samples were assayed in quadruplicate for each time point and condition.

**In vitro invasion assay**. Media containing chemotactic factors (Complete Media) was plated in the bottom of a 24-well plate (BD Falcon) and Boyden chambers (8 μm pore, BD Falcon) pre-coated at 37 °C for 1 h with DMEM containing 5% growth factor-reduced Matrigel (VWR) were placed on top of the complete median in a 24-well plate. A total of $2 \times 10^5$ mouse cells or $2 \times 10^5$ human breast cancer cells were re-suspended in DMEM in the absence of chemotactic factors and added to the upper level of the Boyden chambers. Plates were incubated at 37 °C for 20 h after which cells were fixed in a solution of 10% neutral-buffered formalin for 20 min. Boyden chambers were counterstained with Crystal Violet solution (Sigma) for 20 to 30 min and cells that remained on the upper level of the Boyden chambers were manually removed and chambers were dried overnight. Three representative images of each Boyden chamber were taken and positive-pixel area was calculated using the ImageJ software. Experiments were performed in triplicate and the average values are reported (±S.E.M.).

**Immunofluorescence and immunohistochemistry**. Tumours or mammary glands were fixed in 10% neutral-buffered formalin overnight and embedded in paraffin for sectioning. Sections were cut at 4 μm, de-paraffinized in xylene and antigen retrieval was performed with 10 mM Citrate Buffer (pH 6) using a pressure cooker. Sections were then blocked with 10% Power Block (BioGenex) in PBS for 10 min at room temperature. Sections were incubated with primary antibody at 4 °C overnight. For immunofluorescence, sections were incubated with secondary antibodies (Invitrogen) for 1 h at room temperature, followed by DAPI for 15 min, washed three times in PBS and mounted in ImmuMount (Thermo Scientific). Immunostained samples were imaged using a Zeiss LSM800 confocal microscope and analysed with ZEN software.

For immunohistochemistry (IHC), sections were de-paraffinized and blocked as above, endogenous peroxidase activity was quenched by incubation in 3% hydrogen peroxide for 20 min. Sections were incubated with primary antibody overnight at 4 °C, washed three times in PBS, and incubated with biotinylated secondary antibodies (Vector Elite). After three further washes in PBS, IHC staining was visualized using the Vectastain ABC kit (Vector Laboratories) according to the manufacturer's instructions. Sections were then counterstained with hematoxylin, dehydrated, and mounted with Clearmount (Invitrogen). Images were acquired using an Aperio-XT slide scanner (Leica Biosystems). Antibodies for imaging were purchased from the following vendors and used as the specified concentrations: Ezh2 (clone D2C9, Cell Signalling, 1:200), H3K27me3 (clone C36B11, Cell Signalling, 1:400), PyVmT (clone 762, gift from Dr. Steven Dilworth, 1:200), CK8 (Fitzgerald, 1:500, 20R-CP004), Pan-CK (Ventana, used directly, 760–2135), Foxc1 (Novus Biological, mouse specific, 1:100, NB100-1268), FOXC1 (Sigma, human specific, 1:100, HPA040670), BrdU (Cell Signalling, 1:200, Cat# 5292) and HER2 (clone 4B5, Ventana, use directly, 790–2991).

**Quantification of metastatic lung lesions**. For the quantification of lung metastases, three 10 μm step sections were cut and stained with hemotoxylin and eosin. In each scanned image, individual lung metastases were visually identified by their distinct cellular morphology, and the average number of metastases per sample was calculated. To quantify the average area per lesion, each lesion was manually outlined and calculated using the Aperio Scanscope software interface (Leica Biosystems) for all three step sections per sample, and then averaged.

**Immunoblotting**. Freshly excised mouse tumour tissue was flash frozen in liquid nitrogen, crushed with a mortar and pestle and allowed to thaw briefly before lysing in ice-cold RIPA buffer (50 mM Tris-HCl pH 7.4, 150 mM sodium Chloride, 1% NP-40, 1% sodium deoxycholate, 0.1% SDS, 2 mM EDTA, 0.5 mM AEBSF, 25 mM β-glycerophosphate, 1 mM sodium orthovanadate and 10 mM sodium fluoride). Cultured cells were also lysed in RIPA buffer. Protein concentration was determined by Bradford Assay (Bio-Rad) and 30 μg of total protein was analysed by

immunoblot using the Odyssey CL-X imaging system (LI-COR Biosciences). Antibodies used include; Ezh2 (clone D2C9, Cell Singalling, 1:1000, Cat#5246), H3 total (Cell Signalling, 1:1000, Cat# 3638), H3K27me3 (clone C36B11, Cell Signalling, 1:1000, Cat# 9733), PyVmT (clone 762, a gift from Dr. Steven Dilworth, 1:2000), Foxc1 (mouse specific, Novus Biologicas, 1:1000), α/β-Tubulin (Cell Signalling, 1:1000, Cat# 2148), β-Actin (Sigma, 1:5000, A5316), Vinculin (Cell Signalling, 1:1000, Cat# 4650), IRDye 800 CW Donkey anti-rabbit, IRDye 680 CW Donkey anti-mouse and IRDye 680 CW Donkey anti-goat (LI-COR Biosciences, 1:10,000, 926–32213, 926–68072).

**RNA isolation, cDNA synthesis and qRT-PCR.** Total RNA was extracted from flash frozen mammary tumour or cell lines using an RNeasy Midi Kit (Qiagen). cDNA was prepared by reverse transcribing isolated RNA using M-Mulv Reverse Transcriptase, Oligo-dT(23VN) and murine RNase inhibitor (New England Biolabs). Real time quantitative PCR was performed using SYBR Green Master Mix (Roche) and Light Cycler 480 instrument. Samples were run in duplicate and normalized to the internal control, β-Actin. Human and mouse specific primers used are defined in Supplementary Table 1.

**Microarray data acquisition and analysis.** Microarray studies using Affymetrix Mouse Gene 2.0 arrays were performed at the McGill University and Genome Quebec Innovation Centre. Total RNA was extracted from five wild-type and five Ezh2−/− Tet-ON PyVmT tumours at tumour burden endpoint and quantified using a NanoDrop Spectrophotometer ND-1000 (NanoDrop Technologies, Inc.). RNA integrity was assessed using a Bioanalyzer 2100 (Aglient Technologies). Sense-strand cDNA was synthesized from 100 ng of total RNA and fragmentation and labelling were performed to produce ssDNA with the GeneChip WT Terminal Labeling Kit according to manufacturer's instructions (Affymetrix). After fragmentation and labelling, 3.5 μg of labelled DNA was hybridized on GeneChip Mouse Gene 2.0 ST arrays (Affymetrix) and incubated at 45 °C in the Genechip Hybridization oven 640 (Affymetrix) for 17 h at 60 rpm. GeneChips were then washed in a GeneChips Fluidics Station 450 (Affymetrix) using Hybridization Wash and Stain kit according to the manufacturer's instructions (Affymetrix) and scanned using a GeneChip Scanner 3000 (Affymetrix). All procedures were performed at the Genome Quebec Innovation Center, McGill University. Raw data were first processed to perform gene-level normalization and quality control using Affymetrix Expression Console software (Affymetrix). Processed data were next subjected to Gene Level Differential Expression Analysis using Affymetrix Transcriptome Analysis Console software. Differentially expressed genes with ANOVA $p < 0.05$ were considered in further analyses.

Ingenuity pathways analysis (IPA) was performed on differentially expressed genes between Ezh2−/− and Ezh2+/+ tumours using IPA software (Ingenuity Systems). Significance of the overrepresented canonical pathways as well as functions and diseases were determined using Fischer's exact test to calculate p-values. GSEA software was used to identify the enrichment of genes in Ezh2-null tumours within pre-defined gene data sets from the Molecular Signature Database (MSigDB). Gene data sets from the c2.cp.reactome.v5.1 were analysed using gene sets containing 15 to 2000 genes and the false discovery rate was estimated using 1000 permutations of the data sets.

Hierarchical clustering of the gene expression profiles of Ezh2+/+ and Ezh2−/− Tet-ON PyVmt tumours ($n = 5$) was generated using the Hierarchical Clustering module and Euclidean distance measure within the GenePattern software [http://software.broadinstitute.org]. The tumour expression data set was first filtered using the parameters $p < 0.05$ and fold change > 1.5 within Affymetrix TAC software. Probes with no mapped gene or probes mapped to the same gene with bi-directional expression were removed.

A survey of upstream transcription factor binding of target genes was performed by inputting the identified gene lists into the EnrichR online tool [www.enrichr.com][45] to survey the database that infer transcription factor regulation via genome-wide ChIP experiments such as ChEA, JASPAR and TRANSFAC, and ENCODE. For de novo and known transcription factor binding in a defined gene set into HOMER v3.18 software package[46].

**Chromatin immunoprecipitation.** Ezh2 and Foxc1 ChIP enrichments in PyVmT cells were quantified by qRT-PCR analysis using specific primers and normalized to the average enrichments obtained with two control primer sets amplifying non-Ezh2 or non-Foxc1-bound genomic regions. ChIP-seq was performed at the McGill University and Genome Quebec Innovation Centre.

For chromatin immunoprecipitation studies, 5 μg of anti-H3K27me3 (Millipore, 17–622) or rabbit anti-IgG antibody (Cell Signalling, Cat# 2729), or 3 μg of anti-Foxc1 (Novus Biologicals) or rabbit anti-IgG antibody was immobilized overnight at 4 °C on 20 μl of Magna ChIP Protein A + G magnetic beads (Millipore) diluted in 250 μl of PBS + 0.5% BSA and then washed three times with PBS + 0.5% BSA. For cell lines, cells in 15 cm plates were fixed with a 1% final concentration of formaldehyde for 10 min at room temperature and then lysed and sonicated. For tumour tissue, 0.5 g of tumour was homogenized using a tissue homogenizer in 1% final concentration of formaldehyde and let to fix for 10 min after which the samples were lysed and sonicated. Equal amounts of chromatin from different treatment conditions or genotypes were diluted in 2.5 × ChIP

dilution buffer (EDTA 2 mM, NaCl 100 mM, Tris 20 mM, Triton 0.5%) + 100 μl of PBS + 0.5% BSA and added to the antibody-bound beads and left to rotate overnight at 4 °C. Next, beads were washed three times for 3 min at 4 °C with 1 ml LiCl buffer (Tris 100 mM, LiCl 500 mM, Na-desoxycholate 1%) then once with 1 ml TE buffer. DNA was eluted with 150 μl of elution buffer (0.1 M NaHCO3, 0.1% SDS) overnight at 65 °C. Chromatin-immunoprecipitated DNA was purified using a QIAquick PCR purification kit (Qiagen) and eluted in 35 μl of elution buffer. Primers used are defined in Supplementary Table 2. FoxC1 primers were designed based on ChIP-seq data from a previous study[47].

**ChIP-Seq analysis.** For each ChIP-sequencing run, five individual ChIPs for each genotype (input and H3K27me3) were pooled following the same protocol described above. A total of 50 ng of DNA was provided per each ChIP-seq to the Génome Québec Innovation Centre for DNA library preparation using the TruSeq DNA sample preparation kit according to Illumina recommendations. The ChIP DNA libraries were sequenced as single 50 bp reads (tags) using the Ilumina Hiseq 2000 sequencer (Illumina). Raw reads were trimmed for length ($n \geq 32$), quality (phred score $\geq 30$) and adaptor sequence using fastx v0.0.13.1. Trimmed reads were (pools of five different tumours per genotype) then aligned to the mouse reference genome mm10 using BWA v0.5.9[48]. Broad peaks were called using MACS v1.4.1 software (mfold = 10,30; bandwith = 300; p-value cutoff = 1E-5) using sequenced libraries of input DNA as control[49]. The fragment length used was the one predicted by the program. Peak list intersections were done using BEDTools v2.12.0[50]. Binding peaks were considered overlapping if at least 1 base of the peaks overlapped. Annotations were obtained using HOMER v3.18[46].

**PDX and histological samples from human patients.** Samples were collected from patients undergoing breast surgeries at the McGill University Health Centre (MUHC) between 1999 and 2015 who provided written, informed consent (MUHC Research Ethics Board protocols SDR-99-780 and SDR-00-966). Transcriptional profiles for the human patients from which histology was used, was previously published[51]. PAM50 classification of samples was performed using the *genefu* R package (version1.16.0)[6] to identify Luminal B patients, and formalin fixed and paraffin-embedded blocks (FFPE) for the appropriate patients was drawn from the archive. Sections were stained with the immunofluorescence protocol earlier described. For quantification of FOXC1 staining, images of 8 different fields for each sample were acquired with a ×20 objective with a LSM 800 confocal microscopy (Leica). 'Image J for Microscopy Software' package for ImageJ software (v1.5, NIH) was the used to quantify the total fluorescence intensity levels of nuclear FOXC1 per sample, as previously reported[52]. Briefly, using ImageJ (v1.48, NIH), mean fluorescence measured, along with several adjacent background readings. The total corrected cellular fluorescence (TCCF) = integrated density −(area of selected cell × mean fluorescence of background readings), was calculated to determine the total fluorescence intensity level of a given channel of interest. Samples were analysed in a blinded fashion, with the technician unaware as to the status of the sample.

**Patient-derived xenografts.** A single fragment of fresh or frozen tumour (~8 mm³), or $1 \times 10^6$ cells in Matrigel, into cleared inguinal mammary fat pads of 3–4-week-old female NOD-SCID mice. Interscapular oestrogen pellets were also subcutaneously implanted in mice transplanted with HCI003 fragments. HCI003 was established as cited previously[30].

**Human data from publicly available data sets.** Transcriptional data for human data sets from The Cancer Genome Atlas (TCGA) was accessed from the UCSC Xena web interface [https://xenabrowser.net/][53]. Kaplan–Meier survival graphs were generated from data available from KM Plotter [www.kmplotter.com][54]. Supplementary Table 3 outlines the probes used. For the protein levels of EZH2 and FOXC1 in different breast cancer patient samples, turkey box plots were generated from published data available through a web interface [http://prot-shiny-vm.broadinstitute.org:3838/BC2016/][29].

**Statistical analysis.** For the comparison between two experimental groups, statistical significance was assessed via Student's t-test, unless stated otherwise. Error bars represent ± SEM.

**Data availability.** Microarray and ChIP-Seq data sets have been deposited in the GEO database at NCBI. All data sets in this study are available under GEO108861 and GSE108899. All data that support the findings of this study are available from the corresponding author upon request. Statistical analyses for all experiments are detailed in the Methods.

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

## Acknowledgements

We would like to acknowledge Cynthia Lavoie, Vasilios Papavasiliou, the Goodman Cancer Research Center Histology Core and the McGill Comparative Medicine & Animal Resources Center for their technical assistance. This work was funded by a Strategic Team Grant funded by the Canadian Institute of Health Research and the Fonds de Recherche du Québec-Santé and the Canadian Institute of Health Research Epigenetics grant (TEC-128089), a DOD CDMRP Breast Cancer Fellowship W81XWH-11-1-0046 (to H.W.S.), the McGill Integrated Cancer Research Training Program (to H.W.S and A. H.), the Systems Biology Program of the Canadian Institutes of Health Research (to A. H.), and a Canada Research Chair in Molecular Oncology (to W.M.). The breast tissue and data bank at McGill University is supported by funding from the Database and Tissue Bank Axis of the Réseau de Recherche en Cancer of the Fonds de Recherche du Québec-Santé and the Québec Breast Cancer Foundation.

## Author contributions

Conceptualization: W.J.M., A.H., H.W.S. Investigation: A.H., D.Z., R.M.J., C.R.D., T.B. Writing of the original draft: A.H., W.J.M. Review and editing: H.W.S., C.R.D., W.J.M. Funding acquisition: W.J.M., H.W.S. Resources: M.P., P.S., A.L.W., N.B., G.B., M.B., V.G.

## Additional information

**Competing interests:** The authors declare no competing interests.

