## [Peer Review File · Nature Communications]

Reviewers' comments:

Reviewer #1 Expert in breast cancer metastasis:

In the manuscript entitled "Targeting EZH2 reactivates a breast cancer subtype-specific antimetastatic transcriptional program", the authors demonstrated that genetic or pharmacological targeting of the Ezh2 inhibits Luminal B breast cancer metastasis in both mouse model and PDX model. Mechanistically, ablation of EZH2 reactivates FOXC1 by elimination of H3K27me3 deposition then FOXC1 drives expression of a panel of anti-metastasis genes. They further found that higher FOXC1 levels are predictive of favorable outcome specifically in Luminal B breast cancer patients and suggested that FOXC1 could serve as a promising therapeutic target for luminal-B subtype breast cancer. Although the manuscript describes interesting findings about the type specific role of EZH2:FOXC1 axis in luminal-B subtype of breast cancer, there are several major concerns that need to be addressed.

Major concerns:

1. The authors stated that EZH2 could serve as a target for Luminal B breast cancer in a type specific manner, while the evidence is not sufficient. First, the authors only used PDX from one Luminal B and one HER2+ patient that may be unrepresentative. Second, this manuscript lacks the experiments of EZH2 ablation on BLBC model. It is reported that EZH2 promotes expression of NF- κ B targets in ER-negative basal-like breast cancer cells (PMID: 21884980) and inhibition of EZH2 by another compound, GSK343, inhibits MDA-MB-231 growth in vitro (25203626). Thus, EZH2 may also be an attractive target for BLBC.

2. The authors assumed that EZH2 exerts its role main through acting as part of PRC2, however, the possibility of role beyond PRC2 should not be neglected. Expand the hierarchical clustering of transcriptomes in Figure 3A by adding EZH2-/- shFoxc1 and Ezh2 +/+ Foxc1 groups may strengthen authors' conclusion.

3. In the first part of Results, the authors claimed that "Interestingly, Ezh2-deficient tumors assessed for H3K27me 3 by immunofluorescence at endpoint exhibited undetectable levels of H3K27me 3 in the epithelium". However, there were still a number of cells with strong H3K27me3 signal. The authors should revise their conclusion according to the results.

4. The authors claim that "a global decline in binding peak events across the genome in EZH-null tumors" and identified 328 genes (only represents 22% of genes that loss of H3K27M3) that are both upregulated in EZH2-KO cells and lack of H3K27M3 (Fig 3B-3C). Among these 328 genes, only 23 of them contain potential FoxC1-binding sites (which is 14% of 328 genes) (Fig 4A-4B). Again, for these 23 genes, only 4 of them are ChIP-positive for FoxC1 binding on these promoters (Fig 4D). This casts a doubt, how representative of FoxC1 in mediating the downstream function of EZH2-KO? Given that the loss of H3K27M3 is genome wide and only FoxC1 and its 4 ChIP-positive target genes can completely substitute EZH2's function on metastasis.

5. The function of FoxC1 in luminal B subtype is completely different from its known function in basal-like breast cancer. Given these family transcription factors are commonly inactivated and retained in cytoplasm by phosphorylation (for example, Akt-mediated FoxO3 phosphorylation), and IF staining on Fig 3G shows that the majority of FoxC1 is localized in the cytoplasm, it seems that most of the FoxC1 in EZH2-KO cells are existed in inactivated form.

6. On Fig 5, it would be more convincing that knockdown of FoxC1 is performed in EZH2-KO cells instead of utilization of GSK126.

7. In the second part of Results, the authors claimed that "Loss of H3K27me 3 due to GSK126 administration demonstrated a modest effect on primary tumor growth rate that did not reach statistical significance (Supplementary Figure 2C, D)". However, the difference between two

groups on tumor volume seems more than modest, and P value of the difference on tumor weight is indicated to be smaller than 0.05 which is statistically significant in most cases. Again, the authors should revise their conclusion according to the results.

Minor issues:

1. In the right panel of Figure 7A, the HER2+ group should be labeled in Red color in order to be consistent with the legend.
2. The description of method on transcription factor motif enrichment analysis was missing in the Method section.
3. I noticed that the authors used an IgG control rather than an input control for the ChIP-seq. It's rare for researchers to use an IgG control because it's usually difficult to obtain sufficient DNA in IgG group and that may be a trouble to get an equal depth to the experiment group. Here I just want to make sure that the authors have made the best choice to get unbiased peaks.
4. If possible, the authors may deposit their microarray data and ChIP seq data on a public accessible data base for convenient sharing.
5. In Figure 3D, the gene body of Foxc1 in Ezh2+/+ group was associated with high level of H3K27me3 modification compared to the Ezh2-/- group. I am curious that is the Foxc1 promoter in its gene body? What's the H3K27me3 distribution of FOXC1 gene with regard to human Luminal B breast cancer cells with or without ablation of EZH2?

Reviewer #2 Expert in the role of EZH2 in breast cancer metastasis:

This is an interesting study demonstrating the role of EZH2 ablation in reducing metastasis of luminal B breast cancers. Although the oncogenic role of EZH2 has been studied in a number of tumor types, the precise functions in the different molecular classifications of breast are poorly understood.

The authors observe that there exists in an EZH2-Foxc1 axis in Luminal B breast cancers that is amenable to inhibition, and that inhibition of EZH2 in their mouse model and a Luminal B PDX is sufficient to reduce metastatic progression. Through RNA and ChIP-Seq analysis, the authors offer that EZH2 represses Foxc1 expression to promote metastasis through its canonical H3K27 trimethylation function, and that re-expression of Foxc1 is sufficient to rescue this effect. They then observe that this effect appears to be unique to the Luminal B, and not other molecular subtypes of breast cancer. Although the study is well executed, there are significant issues that need to be addressed.

1. Conceptually, treatment of PYMT mice with GSK-126 does not recapitulate the neoadjuvant setting as stated in the manuscript. Neoadjuvant chemotherapy is used in large aggressive carcinomas while in the setting of the present study, GSK is used in early tumor development (when tumors become palpable, rather than in advanced tumors, as in the clinical setting).
2. The authors ignore a body of literature showing an effect of EZH2 in accelerating breast tumor initiation in MMTV-neu mice (Gonzalez ME PNAS 2014), which represents another breast cancer subtype.
3. It is not understood why H3K27me3 is tested only by immunofluorescence, and not by immunoblots. Further, the expression levels of histone H3 would provide a good control, but are not shown.
4. The focus exclusively on Foxc1 is not well articulated. The protein expression of Foxc1 and its targets in breast cancers is not shown.
5. A more sophisticated quantitation of number of metastatic lesions throughout the paper (e.g.,

Figs. 1D, 2D) would be useful (perhaps as the authors do in Supplemental Fig 5C). Also, from the methods, it's unclear whether these are micro/macro metastasis.

6. How was primary tumor burden affected with GSK-126 in the Tet-ON PyMT system?

7. Although primary tumor growth rate is (somewhat surprisingly) not statistically different with GSK126 in tumors implanted into FVB mice, overall primary tumor burden is lower.

8. A few studies have recently suggested (PMID: 27041579, 28028927) that FOXC1 suppresses ER- α expression (and thereby confers resistance to tamoxifen) in luminal breast cancers. From a translational perspective, it would be interesting to explore or mention this in the discussion.

9. Although the study has several mouse experiments, there are few experiments showing that Foxc1 manipulation has an effect on an actual Luminal B cell line or PDX-derived line.

10. A second Lum B cell line would strengthen results of supplementary figure 5A. Again, controls showing that the drug is actually functioning in these cell lines (which presumably have different IC50?) are needed.

11. Fig 7C – demonstration that the drug is working. How much reduction in global H3K27me3 is there in these two PDX with GSK-126 – these controls are necessary before we can draw conclusions about the effects of GSK-126 in these tumors.

12. While the mRNA data from publicly available datasets support the hypothesis and the potential clinical relevance of EZH2 and Foxc1 in Luminal B breast cancers, the data on protein levels is very limited (only immunofluorescence on a few cases), and it is not clear how this was quantified.

Minor points:

1. Add molecular weights for western blots (Sup. 1A, Figs. 3E, 5D). Though densitometry is not perfect, some quantitation would be good, especially for seeing how much Foxc1 is de-repressed with GSK-126 in 5A and 5D (tubulin is also higher in GSK-126-treated cells, so it's unclear the extent of change). Also, H3K27me3 panels should be added to these blots to show that the drug is working.

2. The representative Foxc1 OE/GSK-126 invasion images in Fig 5A don't seem to match the percentages.

3. Distance between time points on X axis should be adjusted for some graphs (e.g., Supplementary Fig 5D and 5E distance between 0-3 days is same as distance between 28-35).

Reviewer #1

1. The authors stated that EZH2 could serve as a target for Luminal B breast cancer in a type specific manner, while the evidence is not sufficient. First, the authors only used PDX from one Luminal B and one HER2+ patient that may be unrepresentative.

To address this comment, we have expanded our *in vivo* preclinical trials to include another Luminal B, another HER2+ and an additional 2 TNBC PDXs, for a total of 6 PDXs (Supplementary Data Figure 7 D,E,F). As our data demonstrate, the significant reduction of distal lung metastases is only observed in the GSK-126 treated cohort of Luminal B PDXs and not any other breast cancer subtype.

Second, this manuscript lacks the experiments of EZH2 ablation on BLBC model. It is reported that EZH2 promotes expression of NF- κ B targets in ER-negative basal-like breast cancer cells (PMID: 21884980) and inhibition of EZH2 by another compound, GSK343, inhibits MDA-MB-231 growth in vitro (25203626). Thus, EZH2 may also be an attractive target for BLBC.

There are contrasting reports on the effects of EZH2 ablation or pharmacological depletion of H3K27me³ in TNBC cell lines. Contrary to the studies that the reviewer has highlighted, loss of EZH2 or global H3K27me³ in MDA MB 231 has also been demonstrated to **increase** cellular proliferation by Wassef and colleagues (PMID:26637281). While the *in vitro* data is conflicting, experiments performed in transgenic models and PDXs, including our own data and those published by Bae and colleagues (PMID:25043748) indicate that EZH2 is dispensable in the pathogenesis of TNBC. Consistent with this conclusion we show that GSK-126 did not significantly affect tumour growth or distal metastasis in two independent TNBC derived PDXs (Supplementary Figure 7). These data further support the major conclusion of the manuscript that the potential of PRC2 as a pharmacological target is highly dependent on the breast cancer subtype.

2. The authors assumed that EZH2 exerts its role main through acting as part of PRC2, however, the possibility of role beyond PRC2 should not be neglected. Expand the hierarchical clustering of transcriptomes in Figure 3A by adding EZH2-/- shFoxc1 and Ezh2 +/- Foxc1 groups may strengthen authors' conclusion.

Transcriptomic analysis of Foxc1-silenced cells is an interesting topic for further study but is beyond the scope of this manuscript. If we were to do such an experiment, these data would not be appropriate in Figure 3A since these transcriptome profiles come from transgenic tumours, not cells. In our experience, due to the presence of a complex tumor stroma comparison of gene expression profiles from tumours to lentiviral-transduced cell lines would be problematic.

We acknowledge the possibility that Ezh2 may exert functions beyond its role as a member of PRC2 in the discussion, but we feel that detailed examination of this is also beyond the scope of the current study. Our data do not rule out non-canonical functions of Ezh2 in the models presented, nor do we claim that they do, but in this study we have found that PRC2-dependent suppression of Foxc1 and its downstream targets is critical for the metastasis of Luminal B breast cancer cells and this is the focus of our paper.

3. In the first part of Results, the authors claimed that "Interestingly, Ezh2-deficient tumors assessed for H3K27me³ by immunofluorescence at endpoint exhibited undetectable levels of H3K27me³ in the epithelium". However, there were still a number of cells with strong H3K27me³ signal. The authors should revise their conclusion according to the results.

Cells exhibiting high H3K27me³ levels in these tumor samples are stromal cells, which do not express Cre recombinase in this model system and hence have retained Ezh2. To better

illustrate that the cells that are H3K27me³ positive are stromal cells and not tumour epithelium, we have co stained tumour tissue to detect H3K27me³ and the epithelial marker Cytokeratin 8 (Supplemental Figure 1C) or Ezh2, H3K27me³ and Cre (Supplemental Figure 1D). The results show that epithelial-derived cells (CK8-positive) are universally Ezh2-negative and H3K27me³-low, while stromal (CK8-negative cells) express Ezh2 and retain H3K27me³.

4. The authors claim that “a global decline in binding peak events across the genome in EZH-null tumors” and identified 328 genes (only represents 22% of genes that loss of H3K27M3) that are both upregulated in EZH2-KO cells and lack of H3K27M3 (Fig 3B-3C).

The ChIP-Seq data obviously show a global decline in H3K27 tri-methylation. The incomplete overlap of H3K27me³-modified targets and transcriptionally up-regulated genes is to be expected, since not all PRC2 target genes will be re-expressed due to absence of positive regulators (either the right transcription factors are not expressed or they are not active due to absence of upstream signals) or due to repression by other mechanisms (DNA methylation, H3K9 methylation, HDACs). If the reviewer refers to some of the current literature in which EZH2, SUZ12 or H3K27me³ ChIP targets were overlaid with transcriptional data, our data is in line with the number of targets published by other groups (PMID: 23375371, 27889452).

Among these 328 genes, only 23 of them contain potential FoxC1-binding sites (which is 14% of 328 genes) (Fig 4A-4B). Again, for these 23 genes, only 4 of them are ChIP-positive for FoxC1 binding on these promoters (Fig 4D).

FOXC1 was identified as a significantly over-represented factor regulating the 328 genes in the overlap in terms of the presence of its binding sites in their promoters. It is not reasonable to assume that a single factor would drive expression of all genes in the overlap, nor is this necessary for such a factor to exert significant effects on the phenotype of EZH2-deficient cells. Notably, there may be other genes that are not found in the FOXC1 signatures of the bioinformatic databases we interrogated that could in fact be FOXC1 targets in our models, and hence the degree of overlap between FOXC1 targets and genes with loss of H3K27me³ in our model may be an underestimate. However, we emphasize that our analysis was done in an unbiased manner and the data showing enrichment of FOXC1 targets are statistically significant. The reviewer also assumes that we have tested all 23 genes for FOXC1 binding and found that only 4 were ChIP-positive for FOXC1. This is not true – only 4 genes were tested and 4/4 were ChIP-positive for FOXC1. The ChIP-QPCR analysis of these genes was done to validate previously published findings that FOXC1 binds the promoters of the genes identified through bioinformatic analysis in Figure 4. We feel that testing all 23 target genes is not necessary for this purpose.

This casts a doubt, how representative of FoxC1 in mediating the downstream function of EZH2-KO? Given that the loss of H3K27M3 is genome wide and only FoxC1 and its 4 ChIP-positive target genes can completely substitute EZH2's function on metastasis.

We do not claim that Foxc1 is the only factor mediating the downstream effects of Ezh2 knockout, but we did use multiple bioinformatic methods to interrogate regulatory factors of the Ezh2-silenced genes and Foxc1 was the only factor to be identified by multiple independent methods. It was also the only bioinformatically identified factor for which expression was upregulated in the Ezh2-knockout tumors, and we further demonstrated that Foxc1 is itself a PRC2/Ezh2 target. Thus, any doubt is unfounded, as the data, backed by statistical analysis, show that FOXC1 is an important regulator of genes activated when EZH2 is knocked out (Figures 3-4), and the functional data found in the rest of the paper clearly show that Foxc1 mediates the effects of EZH2 ablation on metastasis.

5. The function of FoxC1 in luminal B subtype is completely different from its known function in

basal-like breast cancer. Given these family transcription factors are commonly inactivated and retained in cytoplasm by phosphorylation (for example, Akt-mediated FoxO3 phosphorylation), and IF staining on Fig 3G shows that the majority of FoxC1 is localized in the cytoplasm, it seems that most of the FoxC1 in EZH2-KO cells are existed in inactivated form.

The subtype-specific functions of both EZH2 and FOXC1 in breast cancer are a central focus of this manuscript, and the reviewer is correct in stating that the metastasis-suppressive function of FOXC1 in Luminal B tumors is markedly different from its role in basal-like tumors. We describe this extensively, particularly in our discussion.

In Figure 3G we clearly show by overlaying Foxc1 immunofluorescent staining with DAPI that FoxC1 is present in the nucleus of Ezh2-deficient tumor cells. Overall, the pattern of FoxC1 staining indicates both a cytoplasmic and a nuclear localization. While this could reflect the activity of some negative regulatory pathways, such as those alluded to by the reviewer, we show that the expression of Foxc1 target genes in EZH2-KO and GSK126-treated cells is elevated and that Foxc1 can be ChIPed on the promoters of these genes, indicating that enough FoxC1 is nuclear to sustain its transcriptional activity in the absence of functional Ezh2.

6. On Fig 5, it would be more convincing that knockdown of FoxC1 is performed in EZH2-KO cells instead of utilization of GSK126.

While we agree that this experiment would be supportive, we don't believe that it is essential. In a clinical scenario, in terms of designing a treatment strategy to reduce the risk of metastasis, chemical inhibitors of EZH2 would be used and so this is the most relevant approach to take for these experiments. Furthermore, establishment of primary cell lines from Tet ON PyVmT tumours *in vitro* is not technically possible, due to silencing of the Tet operator element occurring during culturing which renders the cells unresponsive to doxycycline, or activating mutations in the reverse tetracycline-dependent transactivator (rtTA) gene that allow doxycycline-independent activity (a phenomenon we have observed with all our inducible transgenic tumour models).

7. In the second part of Results, the authors claimed that "Loss of H3K27me 3 due to GSK126 administration demonstrated a modest effect on primary tumor growth rate that did not reach statistical significance (Supplementary Figure 2C, D)". However, the difference between two groups on tumor volume seems more than modest, and P value of the difference on tumor weight is indicated to be smaller than 0.05 which is statistically significant in most cases. Again, the authors should revise their conclusion according to the results.

We have amended our text to better reflect our findings. The effect on tumor growth rate was not significant, although there was a small but significant difference in the tumor mass at the experimental endpoint. When all the data are considered together, it seems extremely unlikely that these minor effects on tumor burden could mediate the dramatic effects on metastasis that we observed when EZH2 was knocked out or inhibited in multiple models. Furthermore, in our experiments using Luminal B PDX models, we also found a significant effect of EZH2 inhibition on metastasis in the absence of any effect on primary tumor outgrowth, further supporting the notion that effects of EZH2 loss-of-function on metastasis are independent of effects on primary tumor burden.

Minor issues:

1. In the right panel of Figure 7A, the HER2+ group should be labeled in Red color in order to be consistent with the legend.

We corrected the labelling in Figure 7A to correspond to the correct colour.

2. The description of method on transcription factor motif enrichment analysis was missing in

the Method section.

We have amended our text to include transcription factor motif enrichment analysis in the Methods section.

3. I noticed that the authors used an IgG control rather than an input control for the ChIP-seq. It's rare for researchers to use an IgG control because it's usually difficult to obtain sufficient DNA in IgG group and that may be a trouble to get an equal depth to the experiment group. Here I just want to make sure that the authors have made the best choice to get unbiased peaks.

We apologize for this error in the text of the materials and methods section, which has now been amended. This should have read input, not IgG for the ChIP-Seq experiments. We used IgG controls for the candidate qPCR ChIP experiments only, but not for the ChIP-Seq.

4. If possible, the authors may deposit their microarray data and ChIP seq data on a public accessible data base for convenient sharing.

We have made all data sets accessible via GEO, and included these accession numbers in our Materials and Methods Section.

5. In Figure 3D, the gene body of Foxc1 in Ezh2+/+ group was associated with high level of H3K27me3 modification compared to the Ezh2-/- group. I am curious that is the Foxc1 promoter in its gene body? What's the H3K27me3 distribution of FOXC1 gene with regard to human Luminal B breast cancer cells with or without ablation of EZH2?

We have modified the Figure 3D in include the upstream region which illustrates methylation of the promoter. Its presence in the gene body can be explained by the spreading of the H3K27me3 mark that is an inbuilt feature of PRC2 due to EED binding to H3K27me³ and stimulating the methyltransferase activity (PMCID PMC3760771, PMC2984210, 18931660).

Reviewer #2

1. Conceptually, treatment of PYMT mice with GSK-126 does not recapitulate the neoadjuvant setting as stated in the manuscript. Neoadjuvant chemotherapy is used in large aggressive carcinomas while in the setting of the present study, GSK is used in early tumor development (when tumors become palpable, rather than in advanced tumors, as in the clinical setting).

We disagree with this understanding of how the mouse model correlates with clinical stages of human breast cancer, and we have amended our text to better explain this. The PyVmT model develops aggressive, invasive tumours very quickly. Tet ON PyVmT mice at 2 weeks post-induction have large adenomas and early adenocarcinomas that may even have started to invade in some cases, and early dissemination is a feature of this model. Likewise, when tumours become physically palpable in orthotopic studies, they are already “advanced”, being of considerable size relative to the body size of the mouse. We posit that the experimental conditions frequently used for therapeutic testing in mice, which allow tumors to reach a very large size (such as 1cm diameter) are not representative of any common clinical situation, since very few women would present with large, bulky tumours of an equivalent magnitude.

Furthermore, while neoadjuvant therapy has in the past been primarily used in tumors of large size, this is no longer the only relevant criterion (PMID 25902916) and the use of neoadjuvant therapy is increasing (PMID 24529560, PMID 29242042, PMID26945566). Among the other factors influencing the decision to undertake neoadjuvant therapy is metastatic dissemination to the lymph nodes or elevated risk of metastatic spread (PMID 24101169), since it is thought that neoadjuvant therapy may be more effective for elimination of disseminated cells and micrometastases compared to the adjuvant setting (PMID 29242041). This is of

particular relevance to our study, since we show that Ezh2 inhibition has an anti-metastatic effect.

2. The authors ignore a body of literature showing an effect of EZH2 in accelerating breast tumor initiation in MMTV-neu mice (Gonzalez ME PNAS 2014), which represents another breast cancer subtype.

We have amended our discussion to include reference to this literature. Perhaps the differences in phenotype can be explained the use of EZH2 over expression in the paper by Gonzalez and colleagues, where MMTV-Ezh2 mice were interbred with MMTV-ErbB2 mice. Since Ezh2 is already overexpressed in ErbB2-driven tumours, overexpressing it even further with the MMTV-driven transgene might perturb the stoichiometry of PRC2 and trigger some PRC2-independent non-canonical effects. Such non-canonical effects are an interesting topic for further investigation but are beyond the scope of the current study. Notably, when we inhibit endogenous EZH2 in ErbB2+ models using pharmacological approaches, we do not see any effects on metastasis, although we did observe decreased primary tumor growth in ERBB2+ PDX models which may be in agreement with the findings of Gonzalez et al.

3. It is not understood why H3K27me3 is tested only by immunofluorescence, and not by immunoblots. Further, the expression levels of histone H3 would provide a good control, but are not shown.

The presence of stromal cells, which have high H3K27me3, makes immunoblotting the less desirable approach for the detection of differences between wild-type and Ezh2 KO tumours. To better illustrate that the cells that are H3K27me³ positive are stromal cells and not tumour epithelium, we have co stained tumour tissue with H3K27me3 and the epithelial marker Cytokeratin 8 (Supplemental Figure 1C) or Ezh2, H3K27me3 and Cre (Supplemental Figure 1D). For the *in vivo* PDX trials where chemical inhibition of Ezh2 was used, we have added immunoblots to establish global loss of H3K27me³ (Supplemental Figure 6H).

4. The focus exclusively on Foxc1 is not well articulated. The protein expression of Foxc1 and its targets in breast cancers is not shown.

We have amended the text in our results to provide further background into the focus on Foxc1. In summary, multiple methods of bioinformatic analysis, when applied to data from the Ezh2-deficient tumors, consistently identified Foxc1 as a candidate factor driving the expression of genes undergoing loss of H3K27me³. The methods used included HOMER motif analysis, Ingenuity Pathway Analysis and GSEA. No other single factor was identified using multiple approaches. It was also striking that Foxc1 itself is overexpressed in Ezh2-deficient tumors due to the fact that it is targeted for silencing by PRC2 and was absent or low in endpoint tumours of our wild-type, Ezh2-proficient mouse model and in the Luminal B subset of human tumours.

TMA and other protein-level experiments have described the levels of FOXC1 protein in breast cancer (PMID: 20406990, 28493031, 4806384, 26565916). We also use protein-level data (immunofluorescence) to examine FOXC1 and EZH2 protein levels in a cohort of Luminal B patients. An expansive study of protein expression of Foxc1 targets across breast cancer subtypes is beyond the scope of this paper and would represent an extreme technical and logistical challenge, in terms of securing adequate amounts of tissue from less prevalent subtypes (as the majority of patients are Luminal A) and also in terms of obtaining and validating antibodies for the proteins corresponding to Foxc1 targets that are usable in immunostaining techniques. We also note that many of the downstream targets of FOXC1 have been demonstrated to be anti-metastatic. However, to further support our findings shown in Figure 6A and B, we have incorporated observations from a publicly accessible data set into Supplemental Figure 6C,D. The authors employed quantitative mass-spectrometry-based proteomic analysis of 105 genomically annotated breast cancers from the TCGA patient cohort.

Interestingly, Luminal B breast cancer samples were found to be the only subtype that showed a trend towards an inverse association between protein levels of EZH2 and FOXC1 (Supplemental Figure 6C). For further reference, in Supplemental Figure 6D we have also included the protein levels of FOXC1 signature components for each corresponding sample, organized by subtype. Such studies are extremely challenging and relatively few patients were analyzed, making statistical comparisons difficult, however it is interesting to observe that in Luminal B samples most FOXC1 targets are down at the protein level compared to other subtypes, while in BLBC samples they are elevated.

5. A more sophisticated quantitation of number of metastatic lesions throughout the paper (e.g., Figs. 1D, 2D) would be useful (perhaps as the authors do in Supplemental Fig 5C). Also, from the methods, it's unclear whether these are micro/macro metastasis.

IHC-based scoring in Supplementary Figure 5 was facilitated with excellent antibodies recognizing human cytokeratins that are not cross-reactive with mouse. Lung sections were analyzed using a digital slide scanner and associated software and were scored in a blinded fashion. To do a similar analysis with lungs from the transgenic mice or the orthotopic/tail vein injection studies (Figures 1D, 2D) would require staining an antigen that clearly differentiates the mouse tumour cells from the mouse lung stroma, which is more difficult. A useful feature of the mouse Tet ON PyV mT model is that it develops well-defined overt or “macro” metastases with a distinctive histopathology that are easily discernible with H&E staining, which is not the case for the PDX models, where micrometastases are more prevalent and the IHC-based scoring is necessary. We have clarified in the text of the methods and in the main text the degree of micro/macrometastasis in these models.

6. How was primary tumor burden affected with GSK-126 in the Tet-ON PyMT system?

As stated in the text, GSK-126 did reduce the primary tumor burden in the Tet-ON PyV mT system. However, the mice still developed mammary lesions and the magnitude of the effect on tumor initiation is very unlikely to explain the very large effect on metastatic burden, especially when these data are considered within the context of the other findings of the paper. The findings from the transgenic tumor model with genetic Ezh2 ablation are analogous – while there was an effect on tumor initiation, there was clearly a much larger effect on metastasis. In Luminal B PDX models, we also observed a significant effect of EZH2 inhibition on metastasis in the absence of any significant effect on primary tumor outgrowth, which supports our conclusions regarding the role of EZH2 in the metastasis of Luminal B tumors. We have clarified the text to better describe the effects of genetic and pharmacological targeting of Ezh2 on primary tumor burden.

7. Although primary tumor growth rate is (somewhat surprisingly) not statistically different with GSK126 in tumors implanted into FVB mice, overall primary tumor burden is lower.

We have amended our text to better reflect our findings. The effect on tumor growth rate was not significant, although there was a small but significant difference in the tumor mass at the experimental endpoint. Discordance between caliper measurements and tumor mass measurements are not unexpected, since measuring tumors with calipers on a live mouse is not as accurate as weighing an excised tumor on an analytical balance. Tumor volume measurements are always estimate, since they require use of a mathematical formula containing assumptions about the shape of the tumor, whereas in practice this can be very irregular.

8. A few studies have recently suggested (PMID: 27041579, 28028927) that FOXC1 suppresses ER-a expression (and thereby confers resistance to tamoxifen) in luminal breast

cancers. From a translational perspective, it would be interesting to explore or mention this in the discussion.

We have modified our text in the discussion to include this point, as it further helps it illustrate the subtype specific nature of FOXC1 in breast cancer.

9. Although the study has several mouse experiments, there are few experiments showing that Foxc1 manipulation has an effect on an actual Luminal B cell line or PDX-derived line.

To address this, we over expressed FOXC1 in two different human Luminal B breast cancer cell lines and observed a significant decrease in invasion upon exogenous expression of FOXC1, as assessed invasion using a Boyden Chamber assay (Supplementary Figure 6B). We have also included additional PDX models including an additional Luminal B PDX, which allowed us to validate our findings on the subtype-specificity of the EZH2-FOXC1 regulatory module.

10. A second Lum B cell line would strengthen results of supplementary figure 5A. Again, controls showing that the drug is actually functioning in these cell lines (which presumably have different IC50?) are needed.

We have included 2 more Luminal B cell lines in our human breast cancer panel in Supplementary Figure 6, and included immunoblots for all cell lines to show that the drugs are working, in Supplementary Figure 6, right panel. A review of the literature demonstrates that the same dose of GSK-126 (1-2 uM) is commonly utilized across many different types of cell lines (PMID: 23051747, PMC4932959, 26898301).

11. Fig 7C – demonstration that the drug is working. How much reduction in global H3K27me3 is there in these two PDX with GSK-126 – these controls are necessary before we can draw conclusions about the effects of GSK-126 in these tumors.

We have addressed this comment by performing immunoblots for the PDX tumours treated with vehicle or GSK-126, displayed in Supplementary Figure 7C,D and E.

12. While the mRNA data from publicly available datasets support the hypothesis and the potential clinical relevance of EZH2 and Foxc1 in Luminal B breast cancers, the data on protein levels is very limited (only immunofluorescence on a few cases), and it is not clear how this was quantified.

The number of samples analysed by immunofluorescence was enough to generate statistically significant data showing a clear anti-correlation between EZH2 and FOXC1 protein levels specifically in Luminal B breast cancer. While it would of course be advantageous to have a larger number of samples, Luminal B is not the most abundant breast cancer subtype and gaining access to samples that have the gene expression analysis to confirm the Luminal B subtype plus matching sections for immunofluorescence is extremely difficult. Nonetheless, the strength of the negative correlation between these two proteins is sufficient to allow detection even with a sample size of 20.

As outlined in the text, Image J was used to quantify the total nuclear intensity of EZH2 or FOXC1 in each sample, using a previously published protocol, referred to in the manuscript text.

Minor points:

1. A) Add molecular weights for western blots (Sup. 1A, Figs. 3E, 5D). B) Though densitometry is not perfect, some quantitation would be good, especially for seeing how much Foxc1 is de-repressed with GSK-126 in 5A and 5D (tubulin is also higher in GSK-126-treated cells, so it's unclear the extent of change).C) Also, H3K27me3 panels should be added to these blots to show that the drug is working.

Since our immunoblots were performed with Licor Odyssey system, we quantified the fluorescence and added it to Supplementary Figure 4 A and B. H3K27me3 and total H3 immunoblots were added to Figure 3A and D. The figure preparation guidelines for this journal do not require that molecular weight markers are included on immunoblots.

2. The representative Foxc1 OE/GSK-126 invasion images in Fig 5A don't seem to match the percentages.

In any set of experiments, there is no guarantee that an image that is perfectly “representative” of an average of several replicates will be obtained. We emphasize that the quantitative data show statistically significant differences in invasion.

Reviewers' comments:

Reviewer #1 (Remarks to the Author):

The authors have addressed most of the comments, however, a significant issue of how representative of Foxc1 in mediating the downstream function of Ezh2-ko remains unaddressed. Although the authors mentioned that Foxc1 is "the only candidate factor that appeared in multiple analyses, and the only candidate significantly upregulated in our Ezh2-/- endpoint tumors" (Fig S3F), this does not rule out the possibility of transcription factors becomes activated in the absence of Ezh2 (Fig S3E-F). For example, bioinformatic analyses indicated that target gene promoters are enriched RXR and Mef2a (and others) binding sites (Fig S3E), this argues that these transcription factors became activated in the absence of Ezh2 (which functions as a co-repressor) despite their mRNA did not change before or after Ezh2 knockout. To make it simple, RXR and Mef2a remain association with their targets, however, they are inactivated because of the presence of Ezh2. Inhibition of Ezh2 activates RXR and Mef2a (loss of repressor), however, the mRNA levels of RXR and Mef2a remain unchanged in the presence or absence of Ezh2. In addition, the in vitro rescue experiment (Fig 5D) in PyVmT cells remains weak and non-conclusive. Using human luminal-B breast cancer cell lines could strengthen this conclusion.

Reviewer #2 (Remarks to the Author):

The authors have responded thoroughly to the reviewer's critiques, and the study is stronger.

Reviewer #1 (Remarks to the Author):

The authors have addressed most of the comments, however, a significant issue of how representative of Foxc1 in mediating the downstream function of Ezh2-ko remains unaddressed.

Although the authors mentioned that Foxc1 is “the only candidate factor that appeared in multiple analyses, and the only candidate significantly upregulated in our Ezh2-/- endpoint tumors” (Fig S3F), this does not rule out the possibility of transcription factors becomes activated in the absence of Ezh2 (Fig S3E-F).

For example, bioinformatic analyses indicated that target gene promoters are enriched RXR and Mef2a (and others) binding sites (Fig S3E), this argues that these transcription factors became activated in the absence of Ezh2 (which functions as a co-repressor) despite their mRNA did not change before or after Ezh2 knockout.

To make it simple, RXR and Mef2a remain association with their targets, however, they are inactivated because of the presence of Ezh2. Inhibition of Ezh2 activates RXR and Mef2a (loss of repressor), however, the mRNA levels of RXR and Mef2a remain unchanged in the presence or absence of Ezh2.

In addition, the in vitro rescue experiment (Fig 5D) in PyVmT cells remains weak and non-conclusive. Using human luminal-B breast cancer cell lines could strengthen this conclusion.

We are pleased that our revisions have satisfied the majority of this reviewer’s comments.

We have attempted to emphasize to the reviewer, both in our previous letter and in the revised manuscript, that we do not claim that all phenotypes caused by Ezh2 knockout are exclusively

due to de-repression of *Foxc1*. To be clear, we do not attempt to “*rule out the possibility of transcription factors becoming activated in the absence of Ezh2*”. Indeed, this is precisely what we predict would happen. However, for the purpose of this manuscript it is not feasible to attempt an in-depth functional analysis of all transcription factors implicated in reactivation of gene expression in *Ezh2*-null cells. This is why we conducted multiple bioinformatic analyses of both ChIP-Seq and gene expression data with a view to choosing a key factor – not “the only” factor - to investigate further. Each mode of computational analysis we applied has its own strengths and weaknesses and it is worth considering that, while *Mef2a* and RXR were identified through HOMER transcription factor motif analysis, *Foxc1* was also identified through analysis of the TRANSFAC/JASPAR, ENCODE ChIP-Seq and ChEA databases, in addition to HOMER. The fact that *Foxc1* has known anti-metastatic targets that are activated in *Ezh2*-deficient tumors (Figure 4) was also compelling since the major phenotype of *Ezh2* inhibition in Luminal B tumors is a metastatic block. Again, we are not claiming that *Foxc1* is the only factor for which activity is opposed by *Ezh2*, but these data are fully consistent with the significant involvement of *Foxc1* in the phenotype of *Ezh2*-deficient tumors. We have inserted a statement on page 9 of the manuscript to clarify that *Foxc1* is not the only factor implicated in transcriptional reactivation.

In terms of the specific example set forth by the reviewer, we would first point out that *Ezh2* is not a co-repressor. In fact, the consensus in the field is that the major mechanism of action of *Ezh2* does not involve direct interaction with any transcription factors, including RXR or *Mef2a*. Rather, *Ezh2* is recruited to chromatin following transient inactivation of transcription by DNA-binding repressors, at which point it induces a more stable mode of transcriptional silencing through H3K27 tri-methylation (Comet et al., 2016; Holoch and Margueron, 2017). The reviewer’s comment about *Ezh2* somehow inhibiting the activity of RXR or *Mef2a* is therefore very difficult to comprehend. The most obvious way that RXR and *Mef2a* could be “activated” by loss of *Ezh2* would be if the genes encoding these factors were themselves targets of PRC2. While this is the case for *Foxc1*, our ChIP-Seq and RNA-Seq data do not support this contention for *Mef2a* or any of the *Rxr* genes. Alternatively, target genes of RXR and/or *Mef2a* could be silenced by PRC2. In this case, RXR and *Mef2a* would not “remain associated” with their targets, as the reviewer supposes, because these genes would be packaged in repressive heterochromatin, with transcription factor binding sites inaccessible, and there is no evidence for pioneer factor activity of either *Mef2a* or RXR. However, we agree in general with the reviewer that loss of *Ezh2* could render transcriptional targets of RXR, *Mef2a* and other factors accessible for activation, as suggested in Supplementary Figure 3. Further analysis of the roles of other factors in transcriptional reactivation in *Ezh2*-deficient Luminal B breast cancer cells is beyond the scope of the present manuscript, but would be a fascinating topic for future study. Given the large amount of data we have generated that implicates *Foxc1* in

suppressing metastasis downstream of Ezh2 ablation, we trust that our decision to focus on this transcription factor for the present study was sound.

Regarding the invasion rescue assays in Figure 5D, we would like to point out that the recovery of invasiveness in PyV mT cells treated with GSK126 where Foxc1 was silenced was statistically significant compared to the GSK126-treated control, without Foxc1 silencing. While rescue did not occur to 100% of the level of invasion seen in control cells (without GSK126 treatment), one must bear in mind that, although our RNAi-based strategy was effective, RNAi is never 100% efficient in its ability to silence gene expression. We therefore dispute the contention that this evidence is "weak and inconclusive". Nonetheless, we have followed the reviewer's suggestion and performed an analogous experiment using human Luminal B breast cancer cell lines (MDA-MB-175 and HCC1500). The data, presented in Supplementary Figure 6C, show that FOXC1 is induced by GSK126 treatment in these cell lines, and that diminishing its expression through using two independent siRNAs induces a statistically significant increase in invasion compared to GSK126-treated control (non-FOXC1-silenced) cells, despite incomplete silencing. Thus, these new data support the major conclusion of our study: FOXC1 is a critical driver of an anti-metastatic program in Luminal B breast cancer which becomes activated upon genetic or pharmacological targeting of EZH2.

Sincerely,

Professor William J. Muller, Ph.D

References

- Comet, I., Riising, E.M., Leblanc, B., and Helin, K. (2016). Maintaining cell identity: PRC2-mediated regulation of transcription and cancer. *Nat Rev Cancer* 16, 803-810.
- Holoch, D., and Margueron, R. (2017). Mechanisms Regulating PRC2 Recruitment and Enzymatic Activity. *Trends in Biochemical Sciences* 42, 531-542.

REVIEWERS' COMMENTS:

Reviewer #1 (Remarks to the Author):

The authors have addressed the concerns raised previously, no further comment for this manuscript.